# WMCopier: Forging Invisible Image Watermarks on Arbitrary Images

**Ziping Dong**[1]    **Chao Shuai**[1]    **Zhongjie Ba**[1,2]*    **Peng Cheng**[1,2]    **Zhan Qin**[1,2]
**Qinglong Wang**[1,2]    **Kui Ren**[1,2]
[1]The State Key Laboratory of Blockchain and Data Security, Zhejiang University
[2]Hangzhou High-Tech Zone (Binjiang) Institute of Blockchain and Data Security
Hangzhou, Zhejiang, China
{dongziping,chaoshuai,zhongjieba,peng_cheng,qinzhan,qinglong.wang,kuiren}@zju.edu.cn

## Abstract

Invisible Image Watermarking is crucial for ensuring content provenance and accountability in generative AI. While Gen-AI providers are increasingly integrating invisible watermarking systems, the robustness of these schemes against forgery attacks remains poorly characterized. This is critical, as forging traceable watermarks onto illicit content leads to false attribution, potentially harming the reputation and legal standing of Gen-AI service providers who are not responsible for the content. In this work, we propose **WMCopier**, an effective watermark forgery attack that operates without requiring any prior knowledge of or access to the target watermarking algorithm. Our approach first models the *target watermark distribution* using an unconditional diffusion model, and then seamlessly embeds the *target watermark* into a non-watermarked image via a shallow inversion process. We also incorporate an iterative optimization procedure that refines the reconstructed image to further trade off the fidelity and forgery efficiency. Experimental results demonstrate that WMCopier effectively deceives both open-source and closed-source watermark systems (e.g., Amazon's system), achieving a significantly higher success rate than existing methods[2]. Additionally, we evaluate the robustness of forged samples and discuss the potential defenses against our attack. Code is available at:
`https://github.com/holdrain/WMCopier`.

## 1    Introduction

As generative models raise concerns about the potential misuse of such technologies for generating misleading or fictitious imagery [1], watermarking techniques have become a key solution for embedding traceable information into generated content, ensuring its provenance [2]. Driven by government initiatives [3], AI companies, including Google and Amazon, are increasingly adopting invisible watermarking techniques for their generated content [4, 5], owing to the benefits of imperceptibility and robustness [6, 7].

However, existing invisible watermark systems are vulnerable to diverse attacks, including detection evasion [8, 9] and forgery [10, 11]. Although the former has received considerable research attention, forgery attacks remain poorly explored. Forgery attacks, where non-watermarked content is falsely detected as watermarked, pose a significant challenge to the reliability of watermarking systems. These attacks maliciously attribute harmful watermarked content to innocent parties, such as Generative AI (Gen-AI) service providers, damaging the reputation of providers [12, 13].

---

*means corresponding author.

[2]We have reported this to Amazon AGI's Responsible AI team and collaborated on developing potential defense strategies. For the official statement from Amazon, see Appendix H.

39th Conference on Neural Information Processing Systems (NeurIPS 2025).

Existing watermark forgery attacks are broadly categorized into two scenarios: the black-box setting and the no-box setting. In the black-box setting, the attacker has partial access to the watermarking system: such as knowledge of the specific watermarking algorithm [14], the ability to obtain paired data (clean images and their watermark versions) via the embedding interface [15, 16], or query access to the watermark detector [14]. However, such black-box access is unrealistic in practice, as the watermark embedding process is typically integrated into the generative service itself, rendering it inaccessible to end users, thus disabling paired data acquisition. Moreover, service providers rarely disclose the specific watermarking algorithms they employ [5]. Therefore, our focus is primarily on the no-box setting, where the attacker has neither knowledge of the watermarking algorithm nor access to its implementation, and only a collection of generated images with unknown watermarking schemes is available. Under this setting, Yang et al. [10] attempt to extract the watermark pattern by computing the mean residual between watermarked images and natural images from ImageNet [17], and then directly adding the estimated pattern to forged images at the pixel level. However, this achieves limited performance because it assumes that the watermark signal remains constant across all images. Moreover, its estimation is further hindered by the domain gap between ImageNet images and the unknown clean counterparts of the watermarked samples.

Inspired by recent work [18–21], demonstrating that diffusion models serve as powerful priors capable of capturing complex data distributions, we ask a more exploratory question:

***Can diffusion models act as copiers for invisible watermarks?***

To be more precise, can we leverage them to copy the underlying watermark signals embedded in watermarked images?

Building on this insight, we propose **WMCopier**, a no-box watermark forgery attack framework tailored for practical adversarial scenarios. In this setting, the attacker has no prior knowledge of the watermarking scheme used by the provider and only has access to watermarked content generated by the Gen-AI service. Specifically, we first train an unconditional diffusion model on watermarked images to capture their underlying distribution. Then, we perform a shallow inversion to map clean images to their latent representations, followed by a denoising process that injects the watermark signal utilizing the trained diffusion model. To further mitigate artifacts introduced during inversion, we propose a refinement procedure that jointly optimizes image quality and alignment with the target watermark distribution.

To evaluate the effectiveness of WMCopier, we perform comprehensive experiments across a range of watermarking schemes, including a closed-source one (Amazon's system). Experimental results demonstrate that our attack achieves a high forgery success rate while preserving excellent visual fidelity. Furthermore, we conduct a comparative robustness analysis between genuine and forged watermarks. Finally, we explore a multi-message defense strategy that provides practical guidance for improving future watermark design and deployment.

Our key contributions are summarized as follows:

- We propose **WMCopier**, the first no-box watermark forgery attack based on diffusion models, which forges watermark signals directly from watermarked images without requiring any knowledge of the watermarking scheme.
- We introduce a shallow inversion strategy and a refinement procedure, which injects the target watermark signal into arbitrary clean images while jointly optimizing image quality and conformity to the watermark distribution.
- Through extensive experiments, we demonstrate that **WMCopier** effectively forges a wide range of watermark schemes, achieving superior forgery success rates and visual fidelity, including on Amazon's deployed watermarking system.
- We explore a potential defense strategy that provides insights to improve future watermarking systems.

## 2 Preliminary

### 2.1 DDIM and DDIM Inversion

**DDIM.** Diffusion models generate data by progressively adding noise in the forward process and then denoising from pure Gaussian noise during the reverse process. The forward diffusion process is

modeled as a Markov chain, where Gaussian noise is gradually added to the data $x_0$ over time. At each time step $t$, the noised sample $x_t$ can be obtained in closed form as:

$$x_t = \sqrt{\alpha_t} x_0 + \sqrt{1 - \alpha_t}\, \epsilon, \quad \epsilon \sim \mathcal{N}(0, \mathbb{I}) \tag{1}$$

where $\alpha_t$ is the noise schedule, and $\epsilon$ is standard Gaussian noise.

DDIM [22] is a deterministic sampling approach for diffusion models, enabling faster sampling and inversion through deterministic trajectory tracing. In DDIM sampling, the denoising process starts from Gaussian noise $x_T \sim \mathcal{N}(0, \mathbb{I})$ and proceeds according to:

$$x_{t-1} = \sqrt{\alpha_{t-1}} \cdot \left( \frac{x_t - \sqrt{1 - \alpha_t} \cdot \epsilon_\theta(x_t, t)}{\sqrt{\alpha_t}} \right) + \sqrt{1 - \alpha_{t-1}} \cdot \epsilon_\theta(x_t, t) \tag{2}$$

for $t = T, T-1, \ldots, 1$, eventually yielding the generated sample $x_0$. Here, $\epsilon_\theta(x_t, t)$ denotes a neural network, which is trained to predict the noise added to $x_0$ at step $t$ during the forward process, by minimizing the following objective:

$$\mathbb{E}_{x_0 \sim p_{\text{data}},\ t \sim \mathcal{U}(1,T),\ \epsilon \sim \mathcal{N}(0,\mathbb{I})} \left[ \left\| \epsilon_\theta(x_t, t) - \epsilon \right\|_2^2 \right]. \tag{3}$$

**DDIM Inversion.** DDIM inversion [23, 22] allows an image $x_0$ to be approximately mapped back to its corresponding latent representation $x_t$ at step $t$ by reversing the sampling trajectory. DDIM inversion has found widespread applications in computer vision, such as image editing [23, 24] and watermarking [25, 26]. We denote this inversion procedure from $x_0$ to $x_t$ as:

$$x_t = \texttt{Inversion}(x_0, t). \tag{4}$$

## 2.2 Invisible Image Watermarking

Invisible image watermarking helps regulators and the public identify AI-generated content and trace harmful outputs (such as NSFW or misleading material) back to the responsible service provider, thus enabling accountability attribution. Specifically, the watermark message inserted by the service provider typically serves as a model identifier [27]. For example, Stability AI embeds the identifier `StableDiffusionV1` by converting it into a bit string and encoding it as a watermark [28]. A list of currently deployed real-world watermarking systems is provided in Table 6 in Appendix B.

Invisible image watermarking typically involves three stages: *embedding*, *extraction*, and *verification*. Given a clean (non-watermarked) image $x \in \mathbb{R}^{H \times W \times 3}$ and a binary watermark message $m \in \{0, 1\}^K$, the embedding process uses an encoder $E$ to produce a watermarked image:

$$x^w = E(x, m).$$

During the extraction stage, a detector $D$ attempts to recover the embedded message from $x^w$:

$$m' = D(x^w).$$

During the verification stage, the extracted message $m'$ is evaluated against the original message $m$ using a verification function $V$, which measures their similarity in terms of *bit accuracy*. An image is considered watermarked if its bit accuracy exceeds a predefined threshold $\rho$, where $\rho$ is typically selected to achieve a desired false positive rate (FPR). For instance, to achieve a FPR below 0.05 for a 40-bit message, $\rho$ should be set to $\frac{26}{40}$, based on a Bernoulli distribution assumption [29]. Formally, the verification function is defined as:

$$V(m, m', \rho) = \begin{cases} \text{Watermarked,} & \text{if Bit-Accuracy}(m, m') \geq \rho; \\ \text{Non-Watermarked,} & \text{otherwise.} \end{cases} \tag{5}$$

## 3 Threat Model

In a watermark forgery attack, the attacker forges the watermark of a service provider onto clean images, including malicious or illegal content. As a result, these forged images may be incorrectly attributed to the service provider, leading to reputation harm and legal ramifications.

**Attacker's Goal.** The attacker aims to produce a *forged watermarked image* $x^f$ that visually resembles a given clean image $x$, yet is detected by detector $D$ as containing a target watermark

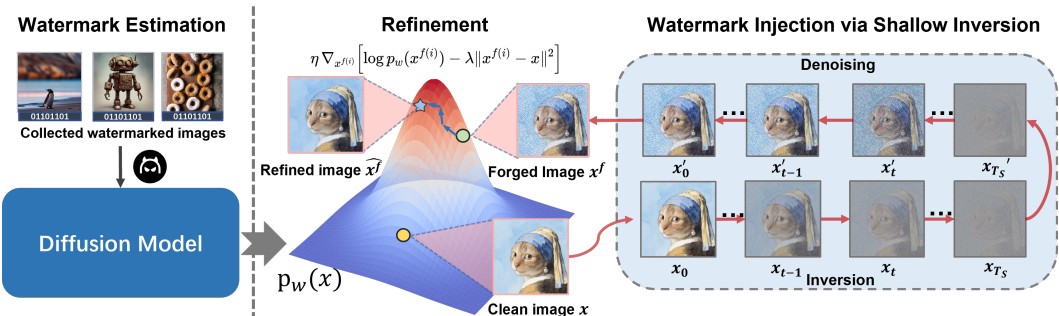

Figure 1: **The pipeline of WMCopier.** The WMCopier consists of three stages. In the first stage, an unconditional diffusion model is trained to estimate the watermark distribution. In the second stage, the estimated watermark is injected into a non-watermarked image using shallow inversion and denoising. Finally, a refinement procedure is applied to mitigate artifacts and ensure conformity to the target watermark distribution $p_w(x)$.

message $m$. Specifically, visual consistency is required to retain the original (possibly harmful) semantic content and to avoid visible artifacts that may reveal the attack.

**Attacker's Capability.** We consider a threat model under the no-box setting:

- The attacker does not know the target watermarking scheme and its internal parameters. They have no access to embed watermarks into their own images and the corresponding detection pipeline.
- The attacker can collect a subset of watermarked images from AI-generated content platforms (e.g., PromptBase [30], PromptHero [31]) or directly use the target Gen-AI service.
- The attacker assumes a static watermarking scheme, i.e., the service provider does not alter the watermarking scheme during the attack period.

## 4  WMCopier

In this section, we introduce **WMCopier**, a watermark forgery attack pipeline consisting of three stages: (1) **Watermark Estimation**, (2) **Watermark Injection**, and (3) **Refinement**. An overview of the proposed framework is illustrated in Figure 1.

### 4.1  Watermark Estimation

Diffusion models are used to fit a plausible data manifold [22, 32, 33] by optimizing Equation 3. The noise predictor $\epsilon_\theta(x_t, t)$ approximates the conditional expectation of the noise:

$$\epsilon_\theta(x_t, t) \approx \mathbb{E}[\epsilon \mid x_t] := \hat{\epsilon}(x_t), \tag{6}$$

which effectively turns $\epsilon_\theta$ into a regressor for the conditional noise distribution.

Now consider a clean image $x$ and its watermarked version $x^w = x + w(w)$, where $w$ denotes the embedded watermark signal, which can also be interpreted as the perturbation introduced by the embedding process. During the forward diffusion process, we have:

$$x_t^w = \sqrt{\alpha_t}(x + w) + \sqrt{1 - \alpha_t}\epsilon = x_t + \sqrt{\alpha_t}w, \tag{7}$$

where $x_t$ is the noisy version of the clean image at step $t$. The presence of the additive term $\sqrt{\alpha_t}w$ implies that the input to the noise predictor carries a watermark-dependent shift. As a result, the predicted noise satisfies:

$$\epsilon_\theta(x_t^w, t) = \hat{\epsilon}(x_t^w) = \hat{\epsilon}(x_t + \sqrt{\alpha_t}w) \approx \hat{\epsilon}(x_t) + \delta_t(w), \tag{8}$$

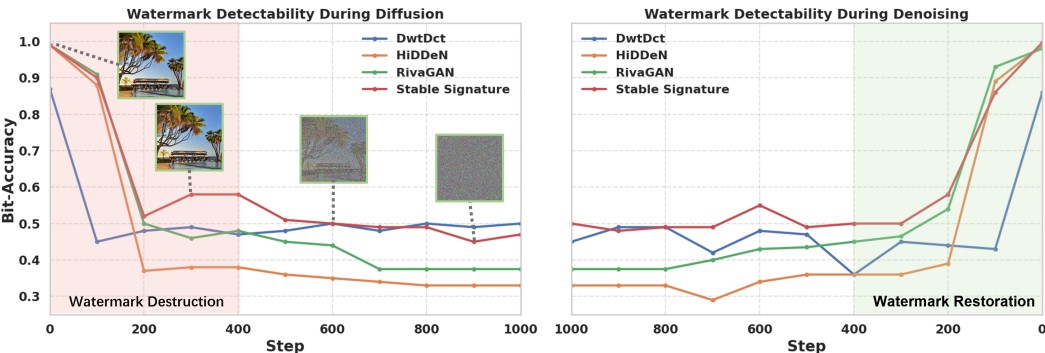

Figure 2: Watermark detectability of four open-source watermarking schemes throughout the diffusion and denoising processes ($T = 1000$). As a reference, the bit accuracy of non-watermarked images remains around 0.5.

where $\delta_t(w)$ denotes the systematic prediction bias introduced by the watermark signal. These biases accumulate subtly at each denoising step, gradually steering the model's output distribution toward the watermarked distribution $p_w(x)$.

To exploit this behavior, we construct an auxiliary dataset $\mathcal{D}_{\text{aux}} = \{x^w | x^w \sim p_w(x)\}$, where each image contains an embedded watermark message $m$. We then train an unconditional diffusion model $\mathcal{M}_\theta$ on $\mathcal{D}_{\text{aux}}$.

Our goal is to obtain forged images $x^f$ with watermark signals while preserving the semantic content of a clean image $x$. Therefore, given the pretrained model $\mathcal{M}_\theta$ and a clean image $x$, we first apply DDIM inversion to obtain a latent representation $x_T$:

$$x_T = \texttt{Inversion}(x, T). \tag{9}$$

The latent representation retains semantic information about the clean image. Starting from $x_T$, we apply the denoising process described in Equation 2 to obtain the forged image $x^f$, where the bias in Equation 8 naturally guides the denoising process toward the distribution of watermarked images.

## 4.2 Watermark Injection

We observe that the reconstructed images with full-step inversion suffer from severe quality degradation, as illustrated in the top row of Figure 3. This phenomenon is attributed to the fact that the inversion of images tends to accumulate reconstruction errors when the input clean images are out of the training data distribution, especially as the inversion depth increases [23, 34, 22]. To mitigate this, we investigate the watermark detectability in watermarked images with four open-source watermarking methods throughout the diffusion and denoising processes. As illustrated in Figure 2, the watermark signal tends to be destroyed gradually during the shallow steps (e.g., $t \leq 400$ for $T = 1000$), Consequently, the watermark signal is restored during these denoising steps.

Therefore, we propose a *shallow inversion* strategy that performs the inversion process up to an early timestep $T_S < T$. By skipping deeper diffusion steps that contribute minimally to watermark injection yet substantially distort image semantics, our method effectively preserves the visual fidelity of reconstructed images while ensuring reliable watermark injection.

## 4.3 Refinement

Although shallow inversion effectively reduces reconstruction errors, forged images may still exhibit minor artifacts (as shown in Figure 3) that cause the forged images to be visually distinguishable, thus exposing the forgery. To address this, we propose a refinement procedure to adjust the forged image $x^f$, defined as:

$$x^{f(i+1)} = x^{f(i)} + \eta \nabla_{x^{f(i)}} \left[ \log p_w(x^{f(i)}) - \lambda \|x^{f(i)} - x\|^2 \right], i \in \{0, 1, ..., L\} \tag{10}$$

where $\eta$ is the step size, $\lambda$ balances semantic fidelity and watermark injection and $L$ is the optimization iterations. The log-likelihood $\log p_w(x^f)$ constrains the samples to lie in regions of high probability

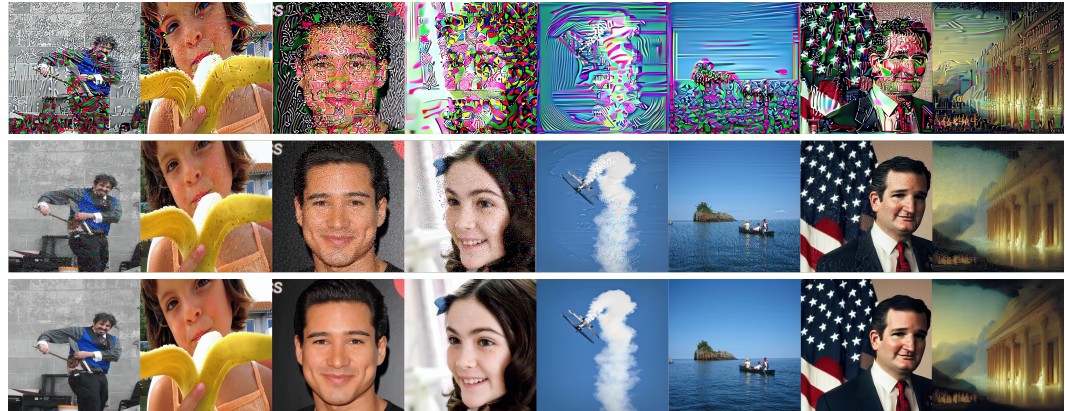

Figure 3: Forged samples generated using full-step inversion, shallow inversion, and shallow inversion with refinement. The first row shows results from full-step inversion ($T_S = T = 100$), where the semantic content of the original clean image is heavily disrupted. The second row corresponds to shallow inversion ($T_S = 40, T = 100$), which introduces only slight artifacts. The third row demonstrates shallow inversion with refinement, where these artifacts are further reduced.

under the watermarked image distribution $p_w(x)$, while the mean squared error (MSE) term $\|x^{f(i)} - x\|^2$ ensures that the refined image remains similar to the clean image $x$. Since the distribution $p_w(x)$ and the conditional noise distribution $p_w^t(x_t)$ are nearly identical at a low noise step $t_l$, the score function $\nabla \log p_w(x)$ can be approximated by $\nabla \log p_w^t(x_t)$. This score can be estimated using a pre-trained diffusion model $\mathcal{M}_\theta$ [35, 36], as defined in Equation 11, where $x_t^f = \sqrt{\alpha_t} x^f + \sqrt{1 - \alpha_t} \epsilon$.

$$\nabla_{x^f} \log p_w(x^f) \approx \nabla_{x_{t_l}^f} \log p_w^{t_l}(x_{t_l}^f) \approx -\frac{1}{\sqrt{1 - \alpha_{t_l}}} \epsilon_\theta(x_{t_l}^f, t_l). \tag{11}$$

By performing this refinement for $L$ iterations, we obtain the forged watermarked image $\hat{x}_f$ after the refinement process. This refinement improves both watermark detectability and the image quality of the forged images, as demonstrated in Figure 3 and Table 11. A complete overview of our WMCopier procedure is summarized in Algorithm 1.

## 5 Evaluation

**Datasets.** To simulate real-world watermark forgery scenarios, we train our diffusion model on AI-generated images and apply watermark forgeries to both AI-generated and real photographs. For AI-generated images, we use DiffusionDB [37] that contains a diverse collection of images generated by Stable Diffusion [38]. For real photographs, we adopt three widely-used datasets in computer vision: MS-COCO [39], ImageNet [17], and CelebA-HQ [40].

**Watermarking Schemes.** We evaluate four watermarking schemes: three post-processing methods—DWT-DCT [41], HiDDeN [42], and RivaGAN [43]—an in-processing method, Stable Signature [27], and a close-source watermark system, Amazon [4]. Each watermarking scheme is evaluated using its official default configuration. A comprehensive description of these methods is included in the Appendix C.

**Attack Parameters and Baselines.** For the diffusion model, we adopt DDIM sampling DDIM sampling with a total step $T = 100$ and perform inversion up to step $T_S = 40$. Further details regarding the training of the diffusion model are provided in the Appendix F. For the refinement procedure, we set the trade-off coefficient $\lambda$ as 100, the number of refinement iterations $L$ as 100, a low-noise step $t_l$ in the refinement as 1 and the step size $\eta$ as $1 \times 10^{-4}$ by default. To balance the attack performance and the potential cost of acquiring generated images (*e.g.*, fees from GenAI services), we set the size of the auxiliary dataset $\mathcal{D}_{aux}$ to 5,000 in our main experiments. For comparison, we consider the method by Yang et al. [10] that operates under the same no-box setting as ours, and Wang et al. [16] that assumes a black-box setting with access to paired watermarked and clean images.

| Attacks | | Black Box | | | No-Box | | | No-Box | | |
|---|---|---|---|---|---|---|---|---|---|---|
| | | Wang et al. [16] | | | Yang et al. [10] | | | Ours | | |
| Watermark scheme | Dataset | PSNR↑ | Forged Bit-acc↑ | FPR@$10^{-6}$↑ | PSNR↑ | Forged Bit-acc.↑ | FPR@$10^{-6}$↑ | PSNR↑ | Forged Bit-acc.↑ | FPR@$10^{-6}$↑ |
| DWT-DCT | MS-COCO | 31.33 | 74.32% | 57.20% | 32.87 | 53.08% | 0.50% | 33.69 | 89.19% | 60.20% |
| | CelebAHQ | 32.19 | 81.29% | 50.70% | 32.90 | 53.68% | 0.10% | 35.29 | 89.46% | 53.20% |
| | ImageNet | 30.16 | 79.64% | 55.10% | 32.92 | 51.96% | 0.20% | 33.75 | 88.25% | 55.80% |
| | Diffusiondb | 31.87 | 78.22% | 50.80% | 32.90 | 51.59% | 0.40% | 33.84 | 85.17% | 54.30% |
| HiddeN | MS-COCO | 31.02 | 80.56% | 77.60% | 29.68 | 63.12% | 0.00% | 31.74 | 99.34% | 95.90% |
| | CelebAHQ | 31.57 | 82.28% | 80.20% | 29.79 | 61.52% | 0.00% | 33.12 | 98.08% | 92.50% |
| | ImageNet | 31.24 | 78.61% | 83.90% | 29.78 | 62.66% | 0.00% | 31.76 | 98.99% | 94.30% |
| | Diffusiondb | 30.74 | 79.99% | 79.20% | 29.68 | 63.36% | 0.00% | 31.46 | 98.83% | 94.60% |
| RivaGAN | MS-COCO | 32.94 | 93.26% | 88.80% | 29.12 | 50.80% | 0.00% | 34.07 | 95.74% | 90.90% |
| | CelebAHQ | 32.64 | 93.67% | 93.80% | 29.23 | 52.29% | 0.00% | 35.28 | 98.61% | 96.00% |
| | ImageNet | 33.11 | 90.94% | 71.40% | 29.22 | 50.92% | 0.00% | 33.87 | 93.83% | 77.10% |
| | Diffusiondb | 33.31 | 89.69% | 80.60% | 29.12 | 48.70% | 0.00% | 34.50 | 90.43% | 84.80% |
| Stable Signature | MS-COCO | 28.87 | 91.68% | 88.90% | 30.77 | 52.67% | 0.00% | 31.29 | 98.04% | 94.60% |
| | CelebAHQ | 32.33 | 79.90% | 90.10% | 30.51 | 51.73% | 0.00% | 30.54 | 96.04% | 100.00% |
| | ImageNet | 29.59 | 85.77% | 85.90% | 30.75 | 51.59% | 0.00% | 31.33 | 97.03% | 98.60% |
| | Diffusiondb | 31.11 | 89.24% | 92.10% | 30.65 | 52.69% | 0.00% | 31.59 | 96.24% | 96.60% |
| Average | | 31.50 | 84.32% | 76.64% | 30.62 | 54.52% | 0.08% | 32.94 | 94.58% | 83.71% |

Table 1: Comparison of our WMCopier with two baselines on four open-source watermarking methods. The cells highlighted in ▢ indicate the highest values in each row for the corresponding metrics. Arrows indicate the desired direction of each metric (↑ for higher values being better).

**Metrics.** We evaluate the visual quality of forged images using Peak Signal-to-Noise Ratio (PSNR), defined as $\mathrm{PSNR}(x, \hat{x^f}) = -10 \cdot \log_{10}\left(\mathrm{MSE}(x, \hat{x^f})\right)$, where $x$ is the clean image and $\hat{x}_f$ is the forged image after the refinement process. A higher PSNR indicates better visual fidelity, *i.e.*, the forged image is more similar to the original. We evaluate the attack effectiveness in terms of bit accuracy and false positive rate (FPR). Bit accuracy measures the proportion of watermark bits in the extracted message that match the target. FPR refers to the rate at which forged samples are incorrectly identified as valid watermarked images. A higher FPR thus indicates a more successful attack. We report FPR at a threshold calibrated to yield a $10^{-6}$ false positive rate on clean images.

### 5.1 Attacks on Open-Source Watermarking Schemes

As shown in Table 5, our WMCopier achieves the highest forged bit accuracy and FPR across all watermarking schemes, even surpassing the baseline in the black-box setting. In terms of visual fidelity, all forged images exhibit a PSNR above 30dB, demonstrating that our WMCopier effectively achieves high image quality. For the frequency-domain watermarking DWT-DCT, the bit accuracy is slightly lower compared to other schemes. We attribute this to the inherent limitations of DWT-DCT, which originally exhibits low bit accuracy on certain images. A detailed analysis is presented in Appendix D.1.

| Watermark Scheme | Attack | Yang et al. [10] | | Ours | |
|---|---|---|---|---|---|
| | Dataset | PSNR↑ | SR↑/Con.↑ | PSNR↑ | SR↑/Con.↑ |
| Amazon WM | Diffusiondb | 23.42 | 29.0%/2 | **32.57** | **100.0%/2.94** |
| | MS-COCO | 24.18 | 32.0%/2 | **32.93** | **100.0%/2.97** |
| | CelebA-HQ | 24.10 | 42.0%/2 | **31.84** | **100.0%/2.98** |
| | ImageNet | 23.95 | 28.0%/2 | **32.88** | **99.0%/2.89** |

Table 2: Performance comparison of baseline and WMCopier on Amazon Watermark.

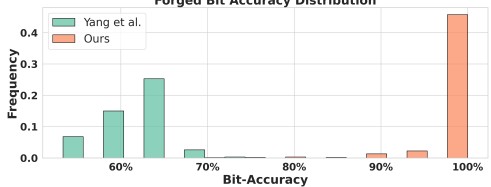

Figure 4: Comparison of forged bit accuracy distribution: Yang's method. vs. Ours.

### 5.2 Attacks on Closed-Source Watermarking Systems

In this subsection, we evaluate the effectiveness of our attack and Yang's method in attacking the Amazon watermarking scheme. The results are shown in Table 2. The success rate (SR), which represents the proportion of images detected as watermarked, and the confidence levels (Con.) returned by the API, are used to evaluate the effectiveness of the attacks on deployed watermarking systems. Compared with Yang's method, our attack achieves superior performance in terms of

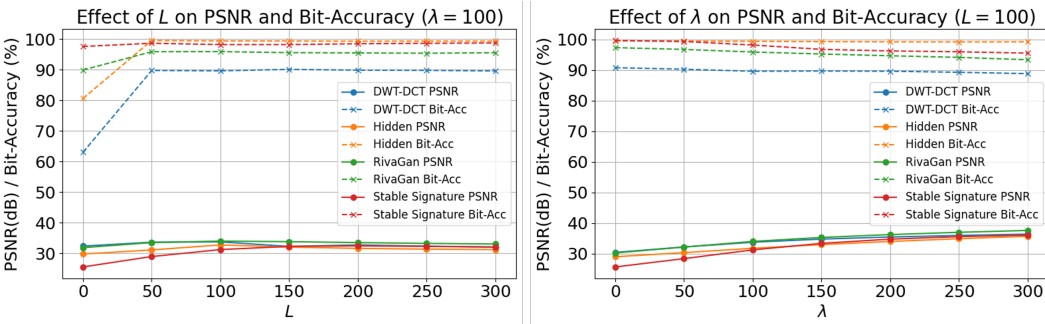

Figure 5: Effect of refinement iterations $L$ (left) and trade off coefficient $\lambda$ (right) on PSNR and Bit-Accuracy under our forgery attacks, with fixed $\eta = 10^{-4}$.

both visual fidelity and forgery effectiveness. Specifically, our method achieves an average PSNR exceeding 30dB and a success rate(SR) close to 100%, whereas Yang's method typically results in PSNR values below 25dB and SR ranging from 28% to 42%.

Furthermore, our forged images generally receive a confidence level of 3—the highest rating defined by Amazon's watermark detection API—while Yang's results consistently remain at level 2. Since Amazon does not disclose the exact computation of the confidence score, we guess that it may correlate with bit accuracy, based on common assumptions [29]. To further investigate this, we analyzed the distribution of forged bit accuracy of both our method and Yang's on a open-source watermarking scheme. As shown in Figure 4, our method achieves over 80% bit accuracy on RivaGan, significantly outperforming Yang's method, which remains below 70%.

## 5.3 Ablation Study

To evaluate the impact of parameter choices on image quality and forgery effectiveness, we conduct two sets of ablation studies by varying (i) the number of refinement optimization steps $L$ and (ii) the trade-off coefficient $\lambda$. As shown in Figure 5, increasing $L$ initially improves both PSNR and forged bit accuracy, with performance saturating beyond $L = 100$. In contrast, larger $\lambda$ values continuously enhance PSNR but lead to a slight degradation in bit accuracy, likely due to over-regularization. While higher PSNR values generally indicate better visual fidelity, we note that visible artefacts may still occur even at elevated PSNR levels. Nevertheless, since an attacker may prioritize forgery success over perceptual quality, we adopt $\lambda = 100$ in our main experiments. The results presented in Table 11 in Appendix E further validate the effectiveness of the refinement process.

## 5.4 Robustness

To investigate the robustness of the forged images, we evaluated its forged bit accuracy of genuine and forged watermarked images under common image distortions, including Gaussian noise, JPEG compression, Gaussian blur, and brightness adjustment. Since the Stable Signature does not support watermark embedding into arbitrary images, we instead report results on generated images. As shown in Table 3, the forged watermark generally exhibits slightly lower robustness compared to the genuine watermark. While some cases show over 20% degradation (highlighted in red), relying on bit accuracy under distortion for separation is inadequate, as it would substantially compromise the true positive rate (TPR), as discussed in Appendix D.3.

# 6 Related Work

## 6.1 Image Watermarking

Image watermarking techniques can generally be categorized into post-processing and in-processing methods, depending on when the watermark is embedded.

**Post-processing methods** embed watermark messages into images after generation. Non-learning-based methods (*e.g.*, LSB [44], DWT-DCT [41, 45]) suffer from poor robustness under common

| Watermark scheme | Distortion Dataset | JPEG | | Blur | | Gaussian Noise | | Brightness | |
|---|---|---|---|---|---|---|---|---|---|
| | | Genuine | Forged | Genuine | Forged | Genuine | Forged | Genuine | Forged |
| DWT-DCT | MS-COCO | 56.44% | 53.00% | 59.84% | 56.56% | 67.86% | 66.90% | 54.66% | 58.36% |
| | CelebAHQ | 55.42% | 53.14% | 63.12% | 58.26% | 64.84% | 66.49% | 53.89% | 57.73% |
| | ImageNet | 56.08% | 52.31% | 59.37% | 54.39% | 68.27% | 67.60% | 54.08% | 57.37% |
| | Diffusiondb | 58.16% | 53.23% | 62.12% | 55.74% | 66.90% | 64.43% | 54.73% | 56.83% |
| HiddeN | MS-COCO | 58.68% | 58.06% | 78.50% | 71.95% | 54.13% | 49.55% | 82.40% | 78.99% |
| | CelebAHQ | 57.05% | 55.07% | 79.83% | 69.07% | 48.94% | 46.02% | 83.63% | 73.21% |
| | ImageNet | 58.86% | 57.83% | 78.20% | 71.34% | 54.10% | 49.57% | 80.95% | 77.40% |
| | Diffusiondb | 58.57% | 57.61% | 79.69% | 72.89% | 54.41% | 50.19% | 81.53% | 77.66% |
| RivaGAN | MS-COCO | 99.44% | 93.32% | 99.60% | 94.99% | 85.71% | 75.00% | 84.51% | 78.81% |
| | CelebAHQ | 99.92% | 97.22% | 99.97% | 98.23% | 85.93% | 74.83% | 84.60% | 79.53% |
| | ImageNet | 98.95% | 92.00% | 99.28% | 93.89% | 84.95% | 74.74% | 82.77% | 77.25% |
| | Diffusiondb | 96.56% | 84.85% | 97.27% | 86.96% | 77.33% | 66.27% | 79.14% | 71.65% |
| StableSignature | MS-COCO | 93.99% | 89.48% | 86.91% | 68.34% | 73.78% | 67.14% | 92.30% | 88.63% |
| | CelebAHQ | | 86.73% | | 65.42% | | 65.33% | | 86.86% |
| | ImageNet | | 87.73% | | 64.88% | | 61.79% | | 91.41% |
| | Diffusiondb | | 85.69% | | 65.45% | | 61.60% | | 87.45% |

Table 3: Bit Accuracy of the genuine watermark and the forged watermark under various image distortions. The distortion parameters are: Gaussian Noise ($\sigma = 0.05$), JPEG (quality=90), Blur (radius=1), and Brightness (factor=6). Cells with ▨ background indicate a degradation gap between 10% and 20%, and cells with ▨ background indicate a degradation gap greater than 20%.

distortions such as compression and noise. Neural network-based approaches mitigate these issues by combining encoder-decoder architectures and adversarial training [42, 46–48]. However, these methods often rely on heavy training and may generalize poorly to unknown attacks.

**In-processing methods** embed watermarks during image generation, either by modifying training data or model weights [19, 49, 29], or by adjusting specific components such as diffusion decoders [27]. Recent trends explore semantic watermarking, which binds messages to generative semantics (*e.g.*, Tree-Ring [50]; Gaussian shading [51]). However, semantic watermarking has not seen real-world deployment [14]. We discuss the effectiveness of our attack on the semantic watermarking in the Appendix D.2.

## 6.2 Watermark Forgery

Kutter et al. [52] first introduced the concept, also known as the watermark copy attack, under the assumption that the watermark signal was a fixed constant. While this assumption was reasonable for early handcrafted watermarking methods, it no longer holds for modern neural network-based schemes. Subsequent studies [53, 16, 54, 14] have investigated watermark forgery under either white-box or black-box settings, where the attacker either has full access to the watermarking model or can embed watermarks into their own images. However, these approaches still rely on strong assumptions that may not hold in realistic deployment scenarios.

In contrast, the no-box setting assumes that only watermarked images are available to the attacker, without access to the model or embedding process. Yang et al. [10] proposed a heuristic method under this setting by estimating the watermark signal through averaging the residuals between watermarked and clean images, and subsequently re-embedding the estimated pattern at the pixel level. This is the scenario we focus on in this work, as it more accurately reflects practical constraints.

## 7 Defense Analysis

To enhance the deployed watermarking system, we suggest modifying the existing watermark system by disrupting the ability of diffusion models to model the watermark distribution effectively. Specifically, we propose a *multi-message strategy* as a simple yet effective countermeasure. Instead of embedding a fixed watermark message, the system randomly selects one from a predefined message pool $m_1, m_2, m_3, \ldots, m_K$ for each image. During detection, the detector verifies the presence of any valid message in the pool. This strategy introduces uncertainty into the watermark signal, increasing the entropy of possible watermark patterns and making it substantially more difficult for generative models to learn consistent features necessary for forgery. We implement this defense using different message pool sizes ($K = 10, 50, 100$) and test on 100 images for simplicity.

As shown in the Table 4, increasing the value of $K$ leads to the FPR drops to 0% at $K = 50$ and $K = 100$. We further strengthen our attack by collecting more watermarked images. Specifically, we collect 5,000, 20,000, and 50,000 watermarked samples to evaluate the effect of data volume on this defense. As shown in Table 12, the FPR remained consistently at 0% even as the size of $D_{\mathrm{aux}}$ increased. Therefore, embedding multiple messages proves to be a simple yet effective countermeasure against our attack.

Table 4: Performance comparison across different $K$ values.

| Dataset | K=10 | | | K=50 | | | K=100 | | |
|---|---|---|---|---|---|---|---|---|---|
| | PSNR↑ | Forged Bit-acc.↑ | FPR@$10^{-6}$↑ | PSNR↑ | Forged Bit-acc.↑ | FPR@$10^{-6}$↑ | PSNR↑ | Forged Bit-acc.↑ | FPR@$10^{-6}$↑ |
| MS-COCO | 34.73 | 81.63% | 34.00% | 34.62 | 69.78% | 0.00% | 34.86 | 71.56% | 0.00% |
| CelebAHQ | 36.13 | 83.41% | 44.00% | 35.89 | 71.00% | 0.00% | 35.87 | 72.91% | 0.00% |
| ImageNet | 34.55 | 79.25% | 25.00% | 34.35 | 70.09% | 0.00% | 34.58 | 71.44% | 0.00% |
| Diffusiondb | 35.14 | 76.28% | 17.00% | 35.10 | 70.66% | 0.00% | 35.40 | 72.28% | 0.00% |

Table 5: Performance comparison across datasets with a larger size of $D_{aux}$ for $K = 100$.

| Dataset | 5000 | | | 20000 | | | 50000 | | |
|---|---|---|---|---|---|---|---|---|---|
| | PSNR↑ | Forged Bit-acc.↑ | FPR@$10^{-6}$↑ | PSNR↑ | Forged Bit-acc.↑ | FPR@$10^{-6}$↑ | PSNR↑ | Forged Bit-acc.↑ | FPR@$10^{-6}$↑ |
| MS-COCO | 34.86 | 71.56% | 0.00% | 34.78 | 71.91% | 0.00% | 30.77 | 71.94% | 0.00% |
| CelebA-HQ | 35.87 | 72.91% | 0.00% | 34.15 | 72.97% | 1.00% | 27.99 | 72.72% | 1.00% |
| ImageNet | 34.58 | 71.44% | 0.00% | 34.57 | 72.56% | 0.00% | 30.47 | 72.19% | 0.00% |
| DiffusionDB | 35.40 | 72.28% | 0.00% | 34.99 | 72.34% | 0.00% | 31.15 | 72.06% | 0.00% |

# 8 Conclusion

We propose WMCopier, a diffusion model-based watermark forgery attack designed for the no-box setting, which leverages the diffusion model to estimate the target watermark distribution and performs shallow inversion to forge watermarks on a specific image. We also introduce a refinement procedure that improves both image quality and forgery effectiveness. Extensive experiments demonstrate that WMCopier achieves state-of-the-art performance on both open-source watermarking and real-world deployed systems. We explore potential defense strategies, a multi-message strategy, offering valuable insights for the future development of AIGC watermarking techniques.

# 9 Acknowledge

We sincerely thank our anonymous reviewers for their valuable feedback and Amazon AGI's Responsible team for their prompt response. This paper is supported in part by the National Key Research and Development Program of China(2021YFB3100300, 2023YFB2904000 and 2023YFB2904001), the National Natural Science Foundation of China(62441238, 62072395, U20A20178, 62172359 and 62472372), the Zhejiang Provincial Natural Science Foundation of China under Grant(LD24F020010), the Key Research and Development Program of Hangzhou City(2024SZD1A27), and the Key R&D Programme of Zhejiang Province(2025C02264).

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

# A Algorithm

---

**Algorithm 1 WMCopier**

---

**Require:** Clean image $x$; Noise predictor $\epsilon_\theta$ of pretrained diffusion model $\mathcal{M}_\theta$; Inversion steps $T_S$; Refinement iterations $L$; Low noise step $t_l$ for refinement; Step size $\eta$; Trade off coefficient $\lambda$.

**Ensure:** Forged watermarked image $\hat{x^f}$

$\quad x_{T_S} \leftarrow \texttt{Inversion}(x, T_S)$ # Obtain noisy latent at step $T_S$ via DDIM inversion

$\quad x'_{T_S} \leftarrow x_{T_S}$ # Initial the start point of sampling

$\quad$ **for** $t = T_S, T_S - 1, \ldots, 1$ **do** # DDIM sampling

$\quad\quad \epsilon_t \leftarrow \epsilon_\theta(x'_t, t)$

$\quad\quad x'_{t-1} \leftarrow \sqrt{\alpha_{t-1}} \cdot \left( \frac{x'_t - \sqrt{1-\alpha_t} \cdot \epsilon_t}{\sqrt{\alpha_t}} \right) + \sqrt{1 - \alpha_{t-1}} \cdot \epsilon_t$

$\quad$ **end for**

$\quad x^f \leftarrow x'_0$

$\quad$ **for** $i = 1$ to $L$ **do** # Refinement

$\quad\quad$ Sample $z \sim \mathcal{N}(0, \mathbf{I})$

$\quad\quad x_{t_l}^{f(i)} \leftarrow \sqrt{\alpha_{t_l}} \cdot x^{f(i)} + \sqrt{1 - \alpha_{t_l}} \cdot z$ # Add noise to a low noise step $t_l$

$\quad\quad x^{f(i+1)} \leftarrow x^{f(i)} + \eta \cdot \nabla_{x^{f(i)}} \left( -\frac{1}{\sqrt{1-\alpha_{t_l}}} \cdot \epsilon_\theta(x_{t_l}^{f(i)}, t_l)) - \lambda \cdot \|x^{f(i)} - x\|^2 \right)$

$\quad$ **end for**

$\quad$ **return** $\hat{x^f} \leftarrow x^{f(L)}$

---

# B Real-World Deployment

In line with commitments made to the White House, leading U.S. AI companies that provide generative AI services are implementing watermarking systems to embed watermark information into model-generated content before it is delivered to users [3].

Google introduced SynthID [5], which adds invisible watermarks to both Imagen 3 and Imagen 2 [55]. Amazon has deployed invisible watermarks on its Titan image generator [4].

Meanwhile, OpenAI and Microsoft are transitioning from metadata-based watermarking to invisible methods. OpenAI points out that invisible watermarking techniques are superior to the visible genre and metadata methods previously used in DALL-E 2 and DALL-E 3 [6], due to their imperceptibility and robustness to common image manipulations, such as screenshots, compression, and cropping. Microsoft has announced plans to incorporate invisible watermarks into AI-generated images in Bing [7]. Table 6 summarizes watermarking systems deployed in text-to-image models.

Table 6: Watermarking deployment across major Gen-AI service providers.

| Service Provider | Watermark | Generative Model | Deployed | Detector |
|---|---|---|---|---|
| OpenAI | Invisible | DALL·E 2 & DALL·E 3 | In Progress | Unknown |
| Google (SynthID) | Invisible | Imagen 2 & Imagen 3 | Deployed | Not Public |
| Microsoft | Invisible | DALL·E 3 (Bing) | In Progress | Unknown |
| Amazon | Invisible | Titan | Deployed | Public |

# C Watermark Schemes

## C.1 Open-source Watermarking Schemes

**DWT-DCT.** DWT-DCT [41] is a classical watermarking technique that embeds watermark bits into the frequency domain of the image. It first applies the discrete wavelet transform (DWT) to decompose the image into sub-bands and then performs the discrete cosine transform (DCT) on selected sub-bands.

**HiDDeN.** HiDDeN [42] is a neural network-based watermarking framework using an encoder-decoder architecture. A watermark message is embedded into an image via a convolutional encoder, and a

decoder is trained to recover the message. Additionally, a noise simulation layer is inserted between the encoder and decoder to encourage robustness.

**RivaGAN.** RivaGAN embeds watermark messages into video or image frames using a GAN-based architecture. A generator network embeds the watermark into the input image, while a discriminator ensures visual quality.

**Stable Signature.** As an in-processing watermarking technique, Stable Signature [27] couples the watermark message with the parameters of the stable diffusion model. It is an invisible watermarking method proposed by Meta AI, which embeds a unique binary signature into images generated by latent diffusion models (LDMs) through fine-tuning the model's decoder.

**Setup.** In our experiments, all schemes are evaluated under their default configurations, including the default image resolutions (128×128 for HiDDeN, 256×256 for RivaGAN, and 512×512 for both Stable Signature and Amazon), as well as their default watermark lengths (32 bits for DWT-DCT and RivaGAN, 30 bits for HiDDeN, and 48 bits for Stable Signature). With regard to PSNR, we report both the original PSNR of these schemes and the PSNR of our forged samples in Table 7.

Table 7: PSNR of watermarking schemes and our forged samples

| Scheme | DWT-DCT | HiddeN | RivaGAN | Stable Signature |
|---|---|---|---|---|
| PSNR (Original) | 38.50 | 31.88 | 38.61 | 31.83 |
| PSNR (Ours) | 33.69 | 31.74 | 34.07 | 31.29 |

## C.2  Closed-Source Watermarking System

Among the available options, Google does not open its watermark detection mechanisms to users, making it impossible to evaluate the success of our attack. In contrast, Amazon provides access to its watermark detection for the Titan model [56], allowing us to directly measure the performance of our attack. Therefore, we chose Amazon's watermarking scheme for our experiments. Amazon's watermarking scheme, referred to as Amazon WM, ensures that AI-generated content can be traced back to its source. The watermark detection API detect whether an image is generated by the Titan model and provides a confidence level for the detection[3] This confidence level reflects the likelihood that the image contains a valid watermark, as illustrated in Figure 6.

In our experiments, we generated 5,000 images from the Titan model using Amazon Bedrock [57]. Specifically, we used ten different prompts to generate images with the Titan model, which were then employed to carry out our attack. The examples of prompts we used are listed in Figure 7. In this attack, we embedded Amazon's watermark onto four datasets, each containing 100 images. Finally, we submitted the forged images to Amazon's watermark detection API. Additionally, we forged Amazon's watermark on images from non-public datasets, including human-captured photos and web-sourced images, all of which were flagged as Titan-generated.

**Results**

To determine if an image was generated using a Titan Image Generator model, upload an image above and select analyze.

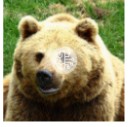

**Watermark detected** (Confidence: High)

Bedrock detected a watermark generated by the Titan Image Generator model ↗ on this image.

Figure 6: Result from Amazon's watermark detection API.

---

[3]Both the Titan model API and the watermark detection service API are accessible via Amazon Bedrock [57].

```
1. A serene landscape of a misty forest at sunrise, with golden light filtering through the trees and a
   calm river flowing in the foreground, ultra-realistic and soft lighting.

2. A futuristic cityscape at night, with glowing neon lights reflecting on wet streets, flying cars and
   towering skyscrapers, cyberpunk style, highly detailed.

3. A majestic lion standing proudly on a cliff at sunset, with a dramatic orange sky and rolling hills in
   the background, hyper-realistic, high detail fur texture.

4. An abstract painting of swirling vibrant colors, reminiscent of Van Gogh's 'Starry Night', using bold
   brushstrokes and a mix of blue, yellow, and white.

5. A beautiful, tranquil Japanese garden with a koi pond, cherry blossom trees in full bloom, and a
   traditional tea house, soft sunlight filtering through the branches.

6. A fantasy scene of a dragon flying over a medieval castle, with smoke rising from its nostrils and a
   stormy sky in the background, highly detailed, dark fantasy style.

7. A close-up of a dew-covered spiderweb in the morning, with sunlight sparkling on the droplets, extremely
   detailed, sharp focus on the texture and reflection.

8. A peaceful 1920s Parisian street view, featuring cozy outdoor cafes, charming cobblestone pathways, and
   vintage buildings with intricate architecture.

9. An astronaut standing on the surface of Mars, gazing at the Earth in the distance, with red rocky
   terrain and a clear blue sky, photorealistic, high contrast.

10.A magical winter wonderland with snow-covered trees, a frozen lake reflecting the pale blue sky, and
   soft sunlight peeking through the branches, ultra-realistic and serene.

                                         …
```

Figure 7: Example prompts used for image generation with the Titan model.

# D    External Experiment Results

## D.1    Further Analysis of DWT-DCT Attack Results

We observed that DWT-DCT suffers from low bit-accuracy on certain images, which leads to
unreliable watermark detection and verification. To reflect a more practical scenario, we assume
that the service provider only returns images with high bit accuracy to users to ensure traceability.
Specifically, we select 5,000 images with 100% bit accuracy to construct our auxiliary dataset $\mathcal{D}_{aux}$.
We then apply both the original DWTDCT scheme and our attack to add watermarks to clean images
from four datasets. As shown in Table 8, our method achieves even higher bit-accuracy than the
original watermarking process itself.

Table 8: Comparison of bit accuracy between original DWT-DCT and DWT-DCT (Ours).

| Dataset | DWTDCT-Original | | DWTDCT-WMCopier | |
|---------|-----------------|-----------------|-----------------|-----------------|
| | Bit-acc.↑ | FPR@$10^{-6}$↑ | Bit-acc.↑ | FPR@$10^{-6}$↑ |
| MS-COCO | 82.15% | 56.60% | **89.19%** | **60.20%** |
| CelebA-HQ | 84.70% | 54.70% | **89.46%** | **53.20%** |
| ImageNet | 85.37% | 55.30% | **88.25%** | **55.80%** |
| DiffusionDB | 82.42% | 52.90% | **85.17%** | **54.30%** |

## D.2    Semantic Watermark

Semantic watermarking [50, 51] embeds watermark information that is intrinsically tied to the
semantic content of the image. To further investigate the effectiveness of our attack on semantic
watermarking, we compare it with the forgery attack proposed by Müller et al. [14], which is specifi-
cally designed for semantic watermark schemes. We adopt Treering [50] as the target watermark.
As shown in Table 9, both our method and Müller's achieve a 100% false positive rate (FPR) under
Treering's default threshold of 0.01. However, our method produces significantly higher forgery
quality, with an average PSNR over 30 dB, compared to around 26 dB for Müller's.

We also evaluate Müller's method on a non-semantic watermark, Stable Signature. As summarized in Table 10, Müller's approach fails to attack this type of watermark, while our method maintains a high success rate.

Table 9: Comparison with Müller et al. [14] and our attack on Treering.

| Dataset | Müller et al. [14] | | Ours | |
|---|---|---|---|---|
| | PSNR↑ | FPR@0.01↑ | PSNR↑ | FPR@0.01↑ |
| MS-COCO | 26.14 | 100.00% | **32.72** | 100.00% |
| CelebA-HQ | 25.22 | 100.00% | **31.52** | 100.00% |
| ImageNet | 26.82 | 100.00% | **32.99** | 100.00% |
| DiffusionDB | 25.19 | 100.00% | **32.78** | 100.00% |

Table 10: Comparison with Müller et al. [14] and our attack on Stable Signature.

| Dataset | Müller et al. [14] | | | Ours | | |
|---|---|---|---|---|---|---|
| | PSNR↑ | Forged Bit-Acc.↑ | FPR@$10^{-6}$↑ | PSNR↑ | Forged Bit-Acc.↑ | FPR@$10^{-6}$↑ |
| MS-COCO | 25.66 | 45.70% | 0.00% | 31.29 | 98.04% | 94.60% |
| CelebA-HQ | 24.73 | 51.23% | 0.00% | 30.54 | 96.04% | 100.00% |
| ImageNet | 25.91 | 47.71% | 0.00% | 31.33 | 97.03% | 98.60% |
| DiffusionDB | 26.12 | 48.45% | 0.00% | 31.59 | 96.24% | 96.60% |

### D.3 Discrimination of Forged Watermarks by Robustness Gap

While the robustness gap between genuine and forged watermarks offers a promising direction for detecting forged samples, we find it is insufficient for reliable discrimination. This limitation becomes particularly evident when genuine samples have already been subjected to mild distortions.

In discrimination, samples are classified as forgeries if their bit accuracy falls below a predefined threshold $\kappa$ after applying a single perturbation. Specifically, we apply perturbation $A$ to both genuine and forged watermark images and then distinguish them based on their bit accuracy. However, considering the inherent robustness of the watermarking scheme itself, when genuine watermarked images have already undergone slight perturbation $B$, the bit accuracy values of genuine and forged samples become indistinguishable. For distortion $A$, we use Gaussian noise with $\sigma = 0.05$, while for distortion $B$, Gaussian noise with $\sigma = 0.02$ is applied. The ROC curve and the bit-accuracy distribution for this case are shown in Figure 8.

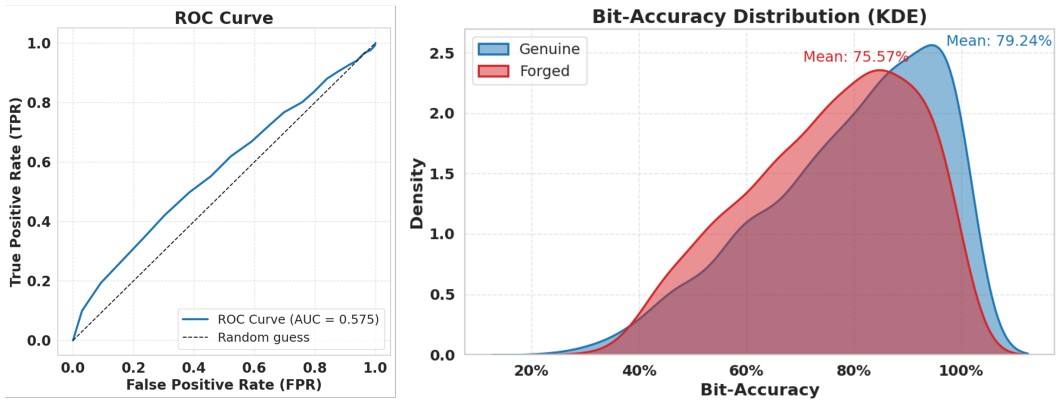

Figure 8: ROC curve and bit accuracy distribution (KDE) for genuine and forged watermark samples under Gaussian noise.

# E    Additional Ablation Studies

Table 11 shows that the proposed refinement step substantially improves visual fidelity, as measured by PSNR, while simultaneously enhancing forgery performance (forged-bit accuracy).

We also explore the impact of varying the size of $D_{\mathrm{aux}}$. Specifically, we use 1,000, 5,000, and 10,000 collected RivaGAN watermarked images. As shown in Table 12, larger $D_{\mathrm{aux}}$ generally yields higher forged-bit accuracy and higher FPR across datasets. However, the improvement becomes marginal once the size of $D_{\mathrm{aux}}$ reaches around 5,000, indicating that the attack performance saturates beyond this point.

Table 11: Impact of refinement on forgery performance.

| Watermark Scheme | PSNR $\uparrow$ | | Forged Bit-acc. $\uparrow$ | | FPR@$10^{-6}$ $\uparrow$ | |
|---|---|---|---|---|---|---|
| | W/o Ref. | W/ Ref. | W/o Ref. | W/ Ref. | W/o Ref. | W/ Ref. |
| DWT-DCT | 32.40 | **33.77** | 63.03% | **89.62%** | 16.00% | **57.00%** |
| HiddeN | 29.81 | **32.79** | 80.60% | **99.40%** | 89.00% | **94.00%** |
| RivaGAN | 31.89 | **34.03** | 89.90% | **95.90%** | 84.00% | **96.00%** |
| StableSignature | 25.60 | **31.27** | 97.58% | **98.19%** | 91.00% | **98.00%** |

Table 12: Performance comparison across datasets with different sizes of $D_{aux}$

| Dataset | 1000 | | | 5000 | | | 10000 | | |
|---|---|---|---|---|---|---|---|---|---|
| | PSNR$\uparrow$ | Forged Bit-acc.$\uparrow$ | FPR@$10^{-6}\uparrow$ | PSNR$\uparrow$ | Forged Bit-acc.$\uparrow$ | FPR@$10^{-6}\uparrow$ | PSNR$\uparrow$ | Forged Bit-acc.$\uparrow$ | FPR@$10^{-6}\uparrow$ |
| MS-COCO | 34.16 | 81.82% | 80.70% | 34.07 | 95.74% | 96.40% | 34.47 | 97.81% | 96.30% |
| CelebA-HQ | 35.74 | 89.10% | 89.50% | 35.28 | 98.61% | 99.10% | 35.25 | 98.63% | 98.50% |
| ImageNet | 34.10 | 81.25% | 71.50% | 33.87 | 93.83% | 94.90% | 34.29 | 93.53% | 95.80% |
| DiffusionDB | 34.77 | 74.76% | 64.10% | 34.50 | 90.43% | 91.20% | 34.96 | 91.70% | 93.60% |

# F    Training Details of the Diffusion Model

We adopt a standard DDIM framework for training, following the official Hugging Face tutorial[4]. The model is trained for 20,000 iterations with a batch size of 256 and a learning rate of $1 \times 10^{-4}$. The entire training process takes roughly 40 A100 GPU hours. To support different watermarking schemes, we only adjust the input resolution of the model to match the input dimensions for each watermark. Other training settings and model configurations remain unchanged. Although the current training setup suffices for watermark forgery, enhancing the model's ability to better capture the watermark signal is left for future work. For our primary experiments, we train an unconditional diffusion model from scratch using 5,000 watermarked images. Due to the limited amount of training data, the diffusion model demonstrates *memorization* [18], resulting in reduced sample diversity, as illustrated in Figure 9. All of the experiments are conducted on an NVIDIA A100 GPU.

# G    Limitation

In this section, we discuss the limitations of our attack. While our current training paradigm already achieves effective watermark forgery, we have not yet systematically explored how to guide diffusion models better to capture the underlying watermark distribution. In this work, we employ a standard diffusion architecture without any specialized training strategies. We leave the exploration of alternative architectures and training schemes to future work. Moreover, understanding why different watermark types exhibit varying forgery and learning behaviors remains an open problem. Additionally, our method requires a substantial amount of data and incurs training costs.

# H    Broader Impact

Invisible watermarking plays a critical role in detecting and holding accountable AI-generated content, making it a solution of significant societal importance. Our research introduces a novel

---

[4]HuggingFace Tutorial:  `https://huggingface.co/docs/diffusers/en/tutorials/basic_tra ining`

watermark forgery attack, revealing the vulnerabilities of current watermarking schemes to such attacks. Although our work involves the watermarking system deployed by Amazon, as responsible researchers, we have worked closely with Amazon's Responsible AI team to develop a solution, which has now been deployed. The Amazon Responsible AI team has issued the following statement:

'On March 28, 2025, we released an update that improves the watermark detection robustness of our image generation foundation models (Titan Image Generator and Amazon Nova Canvas). With this change, we have maintained our existing watermark detection accuracy. No customer action is required. We appreciate the researchers from the State Key Laboratory of Blockchain and Data Security at Zhejiang University for reporting this issue and collaborating with us.'

While our study highlights the potential risks of existing watermarking systems, we believe it plays a positive role in the early stages of their deployment. By providing valuable insights for improving current technologies, our work contributes to enhancing the security and robustness of watermarking systems, ultimately fostering more reliable solutions with a positive societal impact.

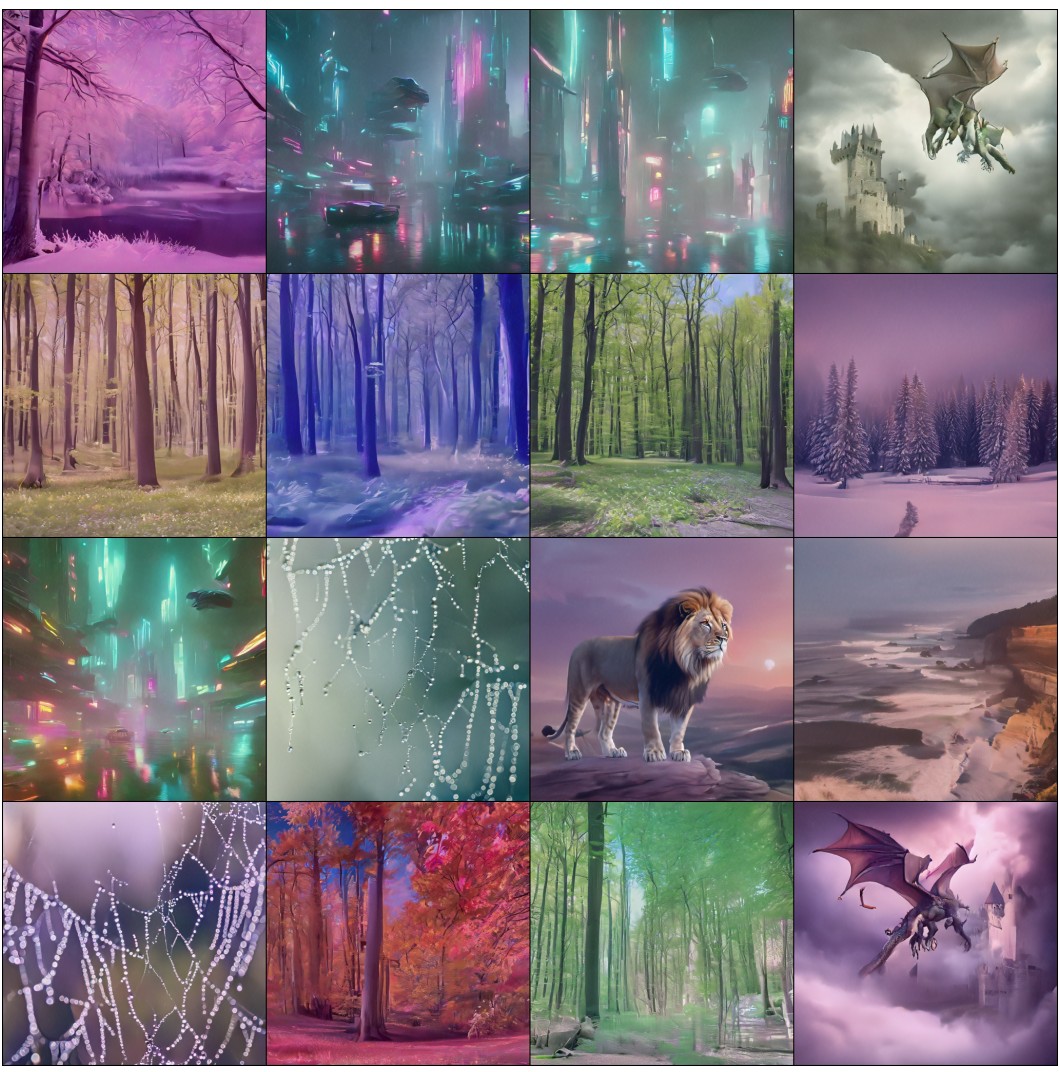

Figure 9: Generated images from diffusion models trained on 5,000 watermarked images

# I Forged Samples

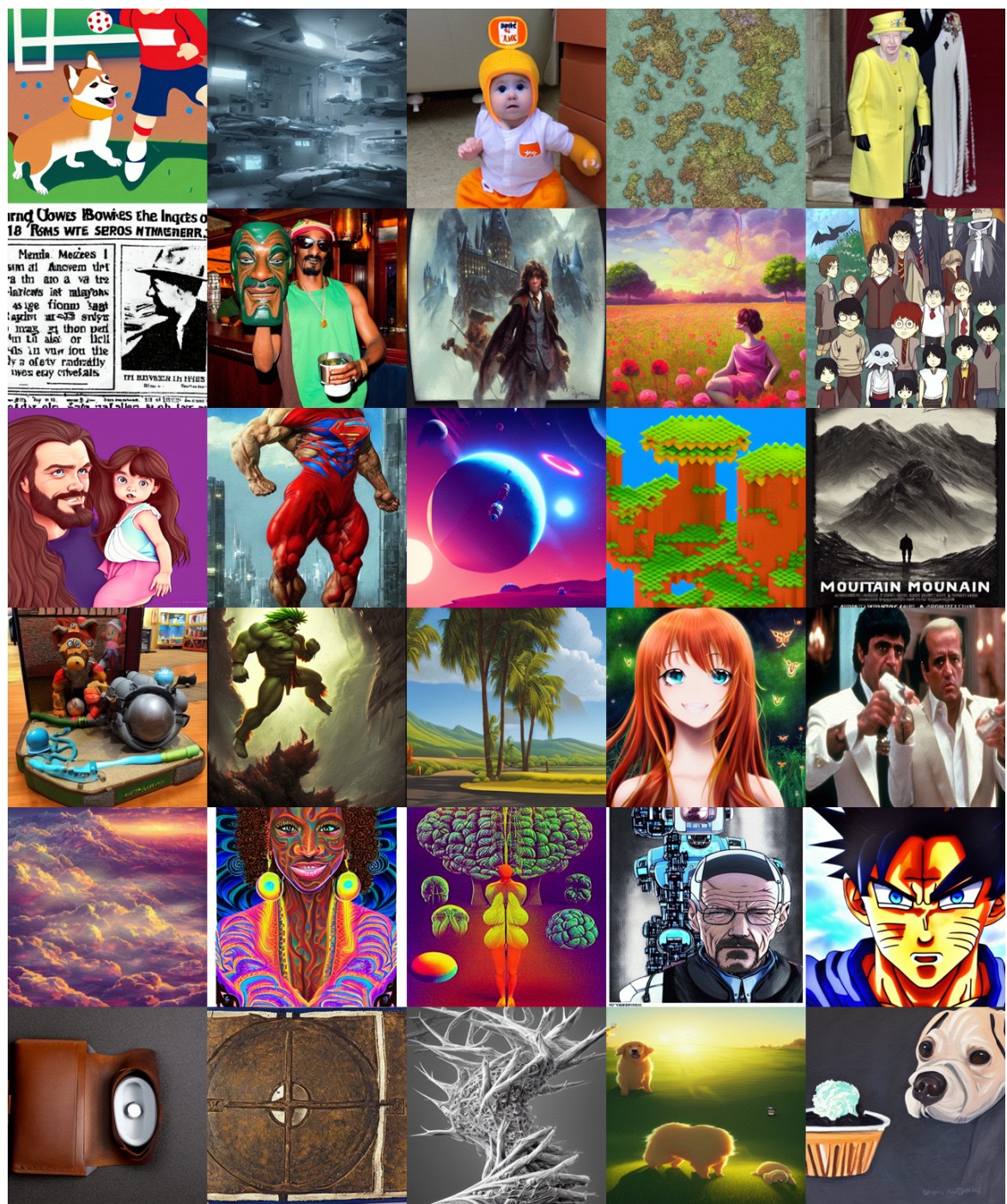

Figure 10: Examples of forged Amazon watermark samples on the DiffusionDB

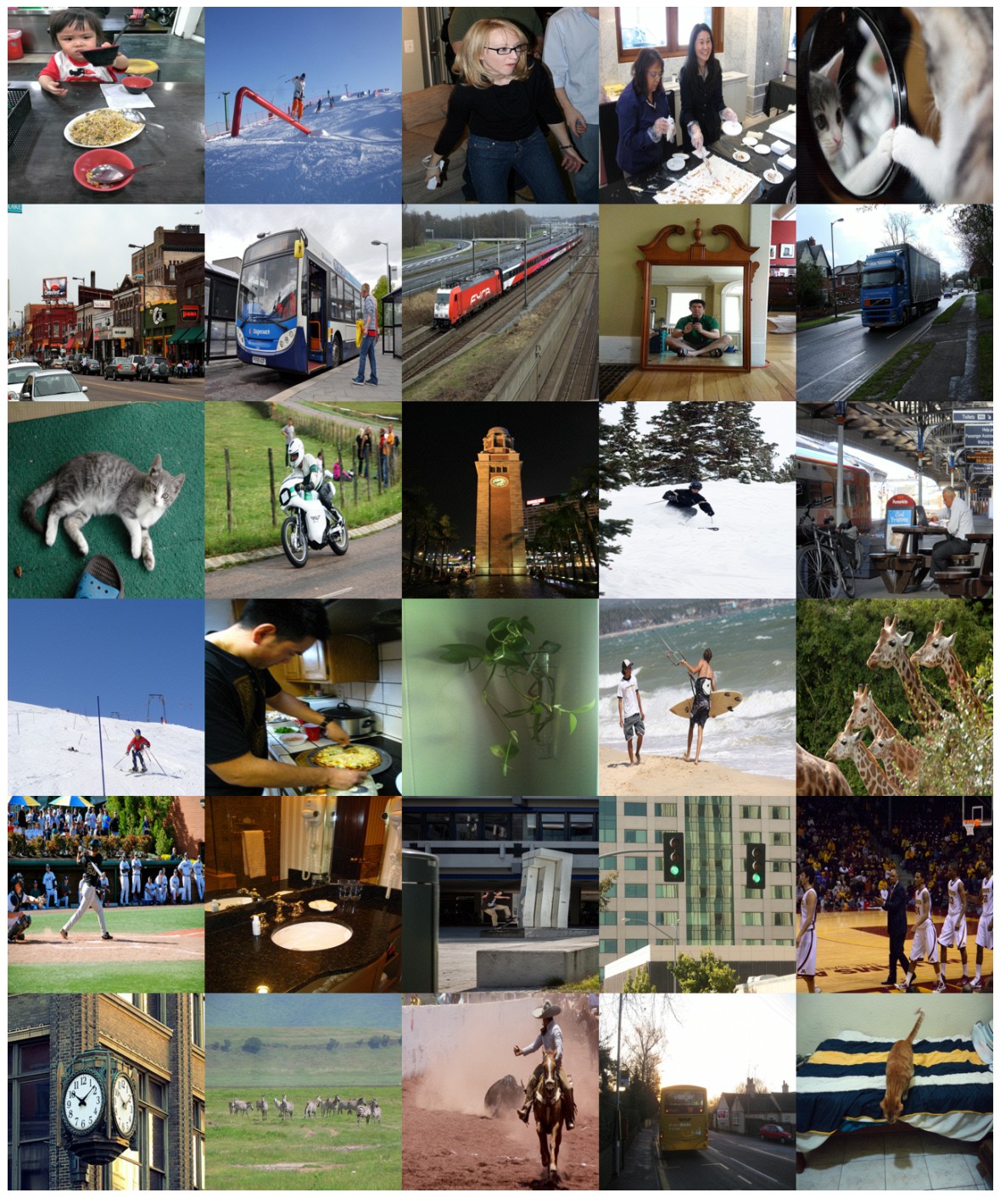

Figure 11: Examples of forged Amazon watermark samples on the MS-COCO

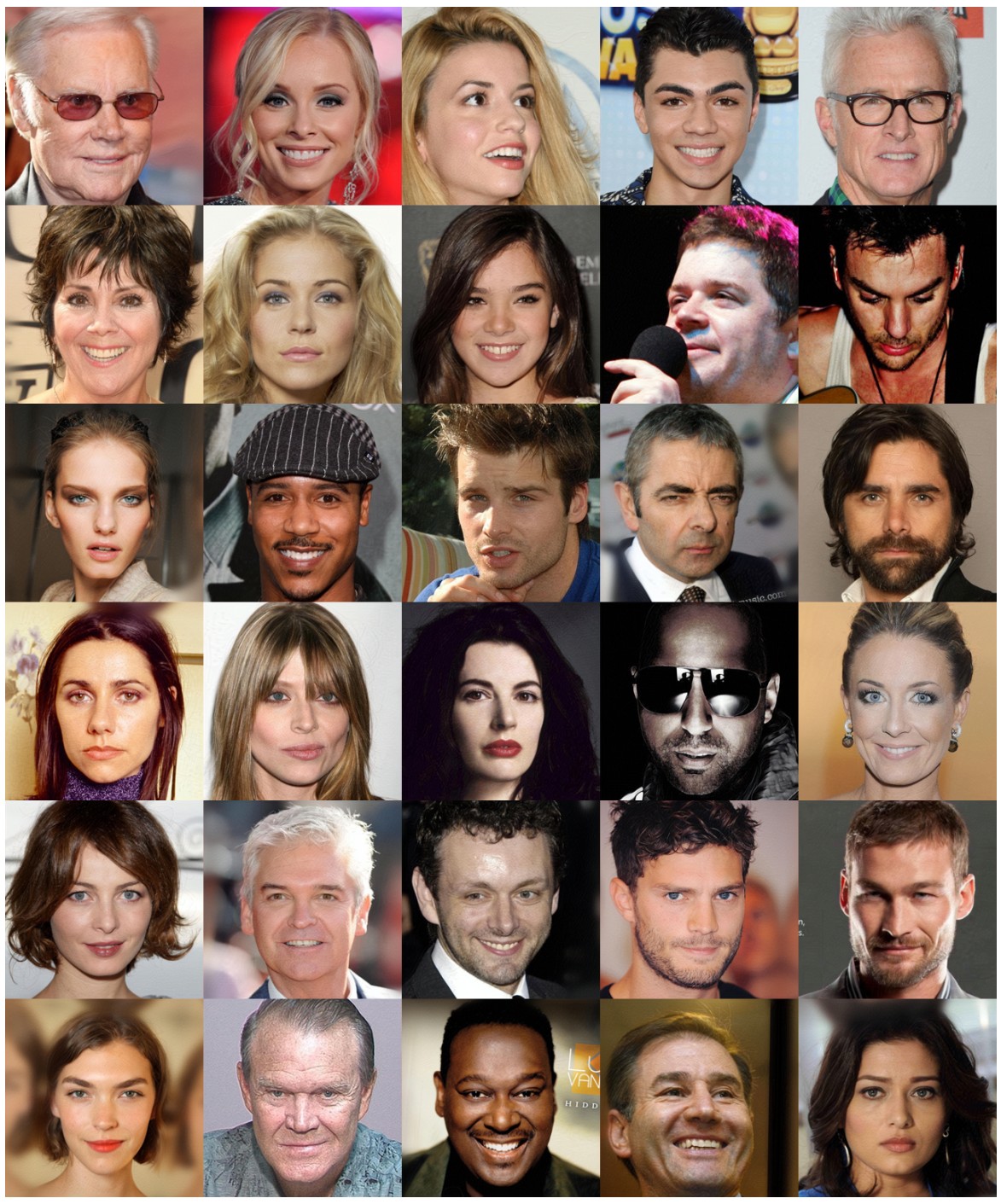

Figure 12: Examples of forged Amazon watermark samples on the CelebA-HQ

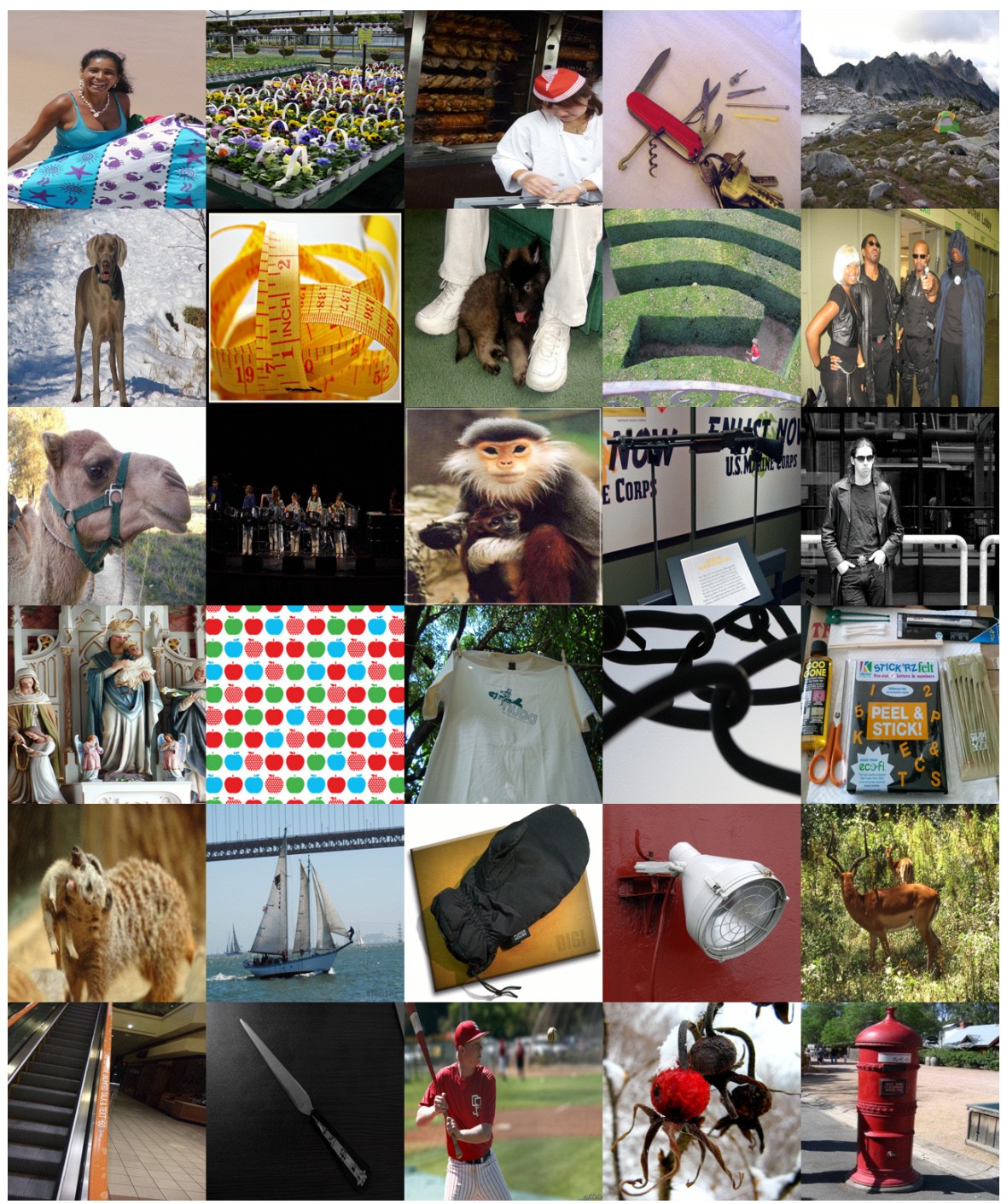

Figure 13: Examples of forged Amazon watermark samples on the ImageNet

