# OpenReview forum: "WMCopier: Forging Invisible Watermarks on Arbitrary Images"
_NeurIPS.cc/2025/Conference — NeurIPS 2025 poster_

### Official Review · Reviewer_aB9i · 2025-06-23

**Clarity:** 2
**Significance:** 3
**Originality:** 2
**Rating:** 5
**Confidence:** 5

**Summary:**

The submission proposes what is known in the watermarking community as a *copy attack*: thanks to the observation of some watermarked contents, the attacker can imitate the watermark in the sense that he can watermark any piece of content illegally.
In the context of GenAI, this is a way to pretend that an original image is indeed AI-generated.

**Questions:**

- What is the impact of the FPR?
- What is the PSNR of the considered watermark embedding?
- How the attack (learning a diffusion process) compares to other alternatives (statistical analysis ou learning a proxy watermark decoder)?
- How the performances of the attack (sucess rate vs. distortion) vary with the number of observed watermarked contents?

**Ethical Concerns:**

["NO or VERY MINOR ethics concerns only"]

**Final Justification:**

After discussing with the authors, and especially reading their list of commitments, I raise my score.

**Limitations:**

Limitations are poorly handled within 5 lines in Appendix H. The main limitation (the attack does not apply to semantic watermarking) is obvious.

**Quality:**

2

**Strengths And Weaknesses:**

# Strengths
1. The paper brings additional proof that new watermarking schemes are not secure.

2. Appendix E.2 is very interesting. It is really a pity that the study gets interesting so late. Investigating defenses against a proposed attack is a sign of soundness.

# Weakness
1. **No reference to watermarking history**

Digital watermarking is an old science (dates back to 1995), so is the art of attacking it. The copy attack was invented in 2000, and since then, defenses and more complex attacks have been proposed. This history is not mentioned (not a single citation).
Please read
https://en.wikipedia.org/wiki/Copy_attack

For instance
   - All the references about watermarking are less than 7 years old, except the worst watermarking papers I know [3] [34].
   - Works on copy attack and defenses proposed in the 2000s are ignored
   - The terminology of watermarking security is ignored. No need for new wordings like "black box" or "no box". The type of attacks (WOA      - Watermark Only Attack / KOA - Known Original Attack / KMA ... ) has been defined a long time ago. There are many papers, surveys, books about watermarking security. Please, use the long-established terminology.

Line 261: "*Classical approaches like LSB, DWT-DCT suffer from poor robustness*" These are not good representatives of classical approaches. Why not compare to *Broken Arrows* (used in international challenges)?

Line 456: the same holds. DWT-DCT is a very poor technique which is not a representative of classical watermarking techniques.

 2. **FPR=5% Is this watermarking?**

The introduction mentions the application of detecting AI-generated content. This implies a very low False Positive Rate, at most $10^{-6}$. FPR=5% is extraordinarily high and absolutely unrealistic. **I strongly disagree** with line 211: "*Following common practice [30], [...] 5% false positive rate*". It turns out that [30] has only 21 citations. I am not sure this paper is a reference for common practices.

The problem is that this choice impacts the experimental results a lot.

For instance, I propose the following attack: Do nothing special. Its distortion is remarkably low (PSNR = $\inf$) and its success rate equals the FPR, ie. already 5%. The higher the FPR, the higher the success rate of any copy-attack.

3. **Attack Distortion**

Several times, the paper claims that the attack "*introduces only slight artifacts*". See caption Fig. 3. , line 216, line 244.
I can clearly see the attack artefacts in Fig. 9, 10, 11, 12 and Fig. 3 last line. Just look at flat, uniform regions, such as the sky.
**I strongly disagree** with line 244:
> Given that PSNR values above 30 dB are generally considered visually indistinguishable to the human eye a PSNR above 30dB means high perceptual quality.

No, typical values in image compression range between 30 and 50dB, where 30dB is for a very poor lossy compression scheme and 50dB is a pristine quality. I say that 40dB is acceptable for watermarking if a perceptual mask based on a Human Visual System model is employed, like Watson, Just-Noticeable-Difference etc.

Moreover, the watermark distortion is not given in the text. Since the authors believe that 30dB is acceptable, perhaps the attack is learned from watermarked images with a high watermarking strength. This facilitates the attack.

4. **Comparison**

There are two alternatives:
- Old attacks are based on averaging (what [47] rediscovered), as well as more elaborate processes such as PCA or ICA. Please read
*Watermarking security: theory and practice*, Teddy Furon, François Cayre, Caroline Fontaine, 2005. This submission only compares with averaging. BTW, I am not convinced by the argument of line 46. The averaging attack is limited because it only works against watermarking schemes where the watermark signal is constant (from one image to another) or almost.
- Learn a proxy watermark detector from the observed watermarked images and run an adversarial example attack to add or remove watermarking. Here are some references:

A transfer attack to image watermarks, arXiv:2403.15365

The Efficacy of Transfer-based No-box Attacks on Image Watermarking: A Pragmatic Analysis, arXiv:2412.02576

5. **Lack of analysis**

As noted in Sect. 4.1, $x^w = x + w$. It would be better to write $x^w = x + w(x)$ for a state-of-the-art (ie. side-informed watermark embedding) zero-bit watermarking scheme. The real question is whether the learned diffusion will capture the dependency of w w.r.t. x.

If the scheme is not SotA and just adds a constant watermark signal (like DCT-DWT), then a simple averaging reveals this secret signal. There is no need for learning a diffusion process. Indeed, it is questionable that the proposed attack does not work so well on such a simple scheme. I also disagree with Line 46. This simple baseline, if well-executed (for instance, by removing low-frequency components before averaging), works well. However, if the watermark signal w depends on x (like in any SotA watermarking scheme) then the averaging makes no sense.

---

> ### Author Rebuttal · Authors · 2025-07-31
>
> We sincerely thank you for your insightful comments and constructive feedback. Your review has helped us better refine and clarify our work.
>
> #### **W1.**
> No reference to watermarking history.
> #### **A1.**
> Thank you for the detailed and insightful feedback.
> We sincerely apologize for overlooking the part of history of classical digital watermarking and the well-established taxonomy used in the community.
> In the revised version, we will update the Related Work section to include classical literature you mentioned and adopt the long-standing terminology.
> Additionally, we will revise inaccurate claims about methods (e.g., LSB, DWT-DCT).
>
> #### **W2&Q1.**
> FPR=5% Is this watermarking? What is the impact of the FPR?
> #### **A2.**
> Thank you for your insightful comment.
> To address the your concern, we have conducted additional experiments under stricter operating points (e.g., FPR = $10^{-6}$) and confirmed that our method maintains strong performance.
> Benefiting from our high bit accuracy, our method maintains a strong forgery success rate even under a stricter evaluation setting (FPR = $10^{-6}$), which requires a much higher number of matching bits for a watermark to be considered valid.
> Specifically, our approach achieves an average success rate of over 80%, while the baseline method by Yang et al. fails almost entirely under the same criterion.
> The results are as follow:
>
> | Watermark Scheme| Dataset| PSNR ↑ (Yang) | Forged Bit-Acc ↑ (Yang) | FPR@10-6 ↑ (Yang) | PSNR ↑ (Ours) | Forged Bit-Acc ↑ (Ours) | FPR@10-6 ↑ (Ours) |
> |-|-|-|-|-|-|-|-|
> |DWT-DCT|MS-COCO|32.87|53.08%|0.50%|33.69|89.19%|60.20%|
> ||CelebAHQ|32.90|53.68%|0.10%|35.29|89.46%|53.20%|
> ||ImageNet|32.92|51.96%|0.20%|33.75|88.25%|55.80%|
> ||Diffusiondb|32.90| 51.59%|0.40%|33.84|85.17%|95.30%|
> |HiddeN| MS-COCO|29.68|63.12%|0.00%|31.74|99.34%|95.90%|
> ||CelebAHQ| 29.79|61.52%|0.00%|33.12|98.08%|92.50%|
> ||ImageNet| 29.78|62.66%|0.00%|31.76|98.99%|94.30%|
> ||Diffusiondb|29.68|63.36%|0.00%|31.46|98.83%|94.60%|
> |RivaGAN| MS-COCO|29.12|50.80%|0.00%|34.07|95.74%|90.90%|
> ||CelebAHQ|29.23|52.29%|0.00%|35.28|98.61%|96.00%|
> ||ImageNet|29.22|50.92%|0.00%|33.87|93.83%|77.10%|
> ||Diffusiondb|29.12|48.70%|0.00%|34.50|90.43%|84.80%|
> |Stable Signature|MS-COCO|30.77|52.67%|0.00%|31.29|98.04%|94.60%|
> ||CelebAHQ|30.51|51.73%|0.00%|30.54|96.04%|100.0%|
> ||ImageNet|30.75|51.59%|0.00%|31.33|97.03%|98.60%|
> ||Diffusiondb|30.65|52.69%|0.00%|31.59|96.24%|96.60%|
> |**Average**||**30.62**|**54.52%**|**0.08%**|**32.94**|**94.58%**|**83.71%**|
>
>
>
> #### **W3&Q3.**
> 1) The watermark distortion is not given in the text. What is the PSNR of the considered watermark embedding?
> 2) I can clearly see the attack artefacts in Fig. 9, 10, 11, 12 and Fig. 3 last line. Just look at flat, uniform regions, such as the sky.
> 3) I strongly disagree with line 244...
>
> #### **A3.**
> Thank you for your comment.
> 1) To clarify, we report the PSNR of the original watermarking schemes and our forgery on the MS‑COCO dataset below. For Stable Signature, we include the PSNR of its generated outputs, as it is not a post-processing method.
>
> |Method|DWT-DCT|HiddeN|RivaGan|Stable Signature|
> |-|-|-|-|-|
> | PSNR (Original)|38.50|31.88|38.61|31.83|
> | PSNR (Ours)|33.69|31.74|34.07|31.29 |
>
> As shown in the table, the PSNR values of our forged images are very close to those of the original watermarking schemes. The largest difference is less than 5 dB.
>
> 2) You also mentioned visible traces in some forged samples from Amazon, especially in background regions such as the sky.
> While our forged images may exhibit slightly lower PSNR compared to the originals, we did not observe obvious visual artifacts in other watermarking schemes.
> To understand this better, we examined the original images watermarked by Amazon.
> We found that similar minor traces also appear in the originals, particularly in low-texture areas like sky or flat color regions.
>
> 3) We will also revise the inappropriate wording in line 244. Thank you for pointing it out.
>
>
> #### **W4&Q3.**
> 1) Old attacks are based on averaging, as well as more elaborate processes such as PCA or ICA in Watermarking security.
> 2) Learn a proxy watermark detector from the observed watermarked images and run an adversarial example attack to add or remove watermarking.
>
> #### **A4.**
> Thank you for your comment.
> 1) Following [1], we used PCA and ICA to estimate the carriers of the watermark signal,
> and then applied an averaging method to forge the RivaGan watermark.
> As the results show, baseline remains difficult to successfully forgery even with techniques such as PCA and ICA.
> We believe this may be because RivaGan does not embed a constant watermark, which aligns with the reason you mentioned.
>
>     |Method|Dataset|PSNR|Forged bit-accuracy|FPR@5%|
>     |-|-|-|-|-|
>     |RivaGAN|MS-COCO|30.23|53.73%|3.20%|
>     ||CelebAHQ|30.16|54.92%|4.10%|
>     ||ImageNet|31.12|50.11%|1.40%|
>     ||Diffusiondb|30.51|48.49%|0.60%|
>
> 2) We reviewed the learing proxy model based method you suggested and attempted to reproduce it.
> This approach [2] involves training 100 proxy watermark models (with different architectures and watermark messages), and then generating adversarial examples against these proxy models, hoping they will generalize to the real watermark model.
> However, this method is designed specifically for watermark removal, adapting this method for watermark forgery (i.e., generating targeted adversarial examples) would be challenging.
> In particular, practical forgery attack assumes no knowledge of the target watermark message, making it difficult to define an appropriate target watermark message for each proxy decoder within their framework.
> As noted in Reference [3], their reproduction of [2] also focused solely on watermark removal rather than forgery.
>
> We also attempted to train a binary classifier to identify ... with architecture Restnet18 and Restnet34 as a proxy watermark detector and perform PGD attack, the results are as follow:
>
> |Proxy Model|Watermark Scheme|PSNR|Forged Bit-Accuracy|FPR@5%|
> |-|-|-|-|-|
> |ResNet-18|RivaGan (MSCOCO)|33.21|47.97%|0.00%|
> |ResNet-34|RivaGan (MSCOCO)|34.19|49.86%|0.00%|
>
> [1] Watermarking security: theory and practice.
>
> [2] A Transfer Attack to Image Watermarks.
>
> [3] The Efficacy of Transfer-based No-box Attacks on Image Watermarking: A Pragmatic Analysis.
>
> #### **W5.**
> 1) As noted in Sect. 4.1, it would be better to write \( x^w = x + w(x) \).
> 2) The argument of line 46 is not convincing... If the scheme is not SotA and just adds a constant watermark signal (like DCT-DWT), then a simple averaging reveals this secret signal.
> 3) This simple baseline, if well-executed (for instance, by removing low-frequency components before averaging), works well.
>
> #### **A5.**
> Thank you for your comment.
> 1) We agree that the expression \( x^w = x + w(x) \) is better, and we will revise our formulation in Section 4.1.
>
> 2) We acknowledge that the statement in line 46 may have been imprecise, and we will revise it accordingly.
> At the same time, we would like to clarify our original intention.
> We believe that using ImageNet samples as substitutes for the clean counterparts is another reason for the limited effectiveness of the baseline method.
> Even if the watermark signal is constant, using the true clean counterparts yields significantly better results compared to using unrelated natural images such as those from ImageNet.
> Additionally, for watermarking schemes such as DWT-DCT, even when the embedded message remains the same, the actual residual added to each image is not constant due to differences in the image content. This explains why the baseline averaging-based attack performs poorly in our setting.
> As you rightly mentioned, while such baselines can be effective on constant watermark signals, our method still demonstrates strong performance on more advanced, learning-based watermarking schemes.
>
> 3) To investigate the effect of removing low-frequency components, we applied Gaussian filtering prior to averaging, using kernel sizes of 7 and 13 with \(\sigma\) values of 1 and 2, respectively, to suppress low-frequency components. The results on the DWT-DCT watermark are reported below.
>
>     | Dataset|PSNR(k=7)|Forged Bit-Acc (k=7)|FPR@5% (k=7)|PSNR (k=13)|Forged Bit-Acc (k=13)|FPR@5% (k=13)|
>     |-|-|-|-|-|-|-|
>     |MS-COCO|36.19|50.64%|2.20%|32.47|50.17%|3.10%|
>     |CelebAHQ|36.24|50.72%|2.20%|32.57|50.12%|2.80%|
>     |ImageNet|36.67|50.64%|2.20%|32.94|49.99%|2.30%|
>     |Diffusiondb|36.13|50.26%|2.30%|32.42|49.77%|2.90%|
>
>     We will continue to explore whether there are better parameter settings to improve the baseline’s performance.
>     If you have any suggestions or insights, we would greatly appreciate your guidance.
>
> #### **Q4.**
> How the performances of the attack (sucess rate vs. distortion) vary with the number of observed watermarked contents?
> #### **A6.**
> Thank you for raising this important point.
> We evaluated our method using watermark datasets of 1,000, 5,000, and 10,000 images, as shown in the results below (with dataset sizes indicated in parentheses).
>
> | Watermark Scheme |Dataset|PSNR(1k)|Forged bitacc.(1k)|FPR@0.05(1k)|PSNR(5k)|Forged bitacc.(5k)|FPR@0.05(5k)|PSNR(10k)|Forged bitacc.(10k)|FPR@0.05(10k)|
> |-|-|-|-|-|-|-|-|-|-|-|
> | RivaGan |MS-COCO|34.16|81.82%|86.80%|34.07|95.74%|100.00%|34.47|97.81%|100.00%|
> |         |CelebAHQ|35.74|89.10%|98.70%|35.28|98.61%|100.00%|35.25|98.63%|99.80%|
> |         |ImageNet|34.10|81.25%|83.70%|33.87|93.83%|100.00%|34.29|93.53%|95.60%|
> |         |Diffusiondb|34.77|74.76%|67.30%|34.50|90.43%|97.40%|34.96|91.70%|98.60%|

---

> > ### Comment · Reviewer_aB9i · 2025-08-03
> >
> > I thank the author for the rebuttal. I have read the other reviews and corresponding rebuttals.
> >
> > > Following [1], we used PCA and ICA to estimate the carriers of the watermark signal, and then applied an averaging method to forge the RivaGan watermark.
> >
> > 1. PCA (or ICA) followed by an average? It does not make sense to me.
> > 2. In any case, it lacks technical details. Did you apply PCA or ICA on raw pixel values? How did you manage such long vectors?
> > 3. More importantly, I might be wrong! Reading this old literature again, it seems that ICA attacks are meaningful when the hidden message varies (and for a spread spectrum like modulation). This is not your case; your message is fixed. I might have wasted your time. I am very sorry for that.
> >
> > I have read the other reviews and the associated rebuttal. I've found it quite interesting.
> >
> > **Can you summarize what you **commit** to modify in the paper and appendices?**
> >
> > There are three things that I will not tolerate (I am sorry to be so direct, but I don't want to be beating around the bush so time is flying):
> >
> > - **FPR = 0.05**. As I said, no application would tolerate such a high FPR. Moreover, since you achieve super high bit accuracy, there is no problem with lowering the FPR. On the contrary, it makes your contribution more shiny. The Table provided in the rebuttal clearly shows that Yang's attack is completely off, whereas yours is still on track. I don't understand why the other tables in the rebuttals are back to 5% FPR. I have the feeling that you want to stick to this unrealistic setting. Please replace all the FPR=5% in the paper with those for a more realistic FPR.
> >
> > - **PSNR values above 30 dB are generally considered visually indistinguishable**. Please don't say this. It's not true. I can see the artefacts. On the contrary, please acknowledge that artefacts are visible but also argue that an attacker might not be as keen on the quality as the "official watermarker."
> >
> > - **The PSNR of the considered watermarking schemes is also too low**. Please acknowledge that this might explain why the attack works so well.  On the other hand, a higher PSNR would certainly imply a larger number of watermarked images (the same effect as increasing $k$). An ablation study on this point would be great.

---

> > > ### Author Response · Authors · 2025-08-04
> > > **Response to Reviewer aB9i and Our Commitments to Modify**
> > >
> > > We sincerely thank you for being such a responsible and constructive reviewer.
> > > Your feedback has been instrumental in helping us improve the quality of our paper, and we are fully committed to addressing your comments in our revision.
> > > Below are our point-by-point responses:
> > >
> > > #### 1. PCA followed by an average?
> > >
> > > We would like to clarify that we apply PCA and ICA directly to the pixel values of the input images to extract their principal components, which serve as an estimation of the underlying cover image. The residual between the original image and this estimation is then computed, and the average of these residuals is used as the extracted watermark signal.
> > > We also truly appreciate the reviewer’s openness and reconsideration. These thoughtful reflections have helped us improve the exposition of our method.
> > >
> > >
> > > #### 2. FPR = 0.05.
> > > Thank you very much for your suggestion.
> > > We fully agree that lowering the FPR is beneficial to our contribution, and we are committed to updating all FPR settings in the revised paper to $10^{-6}$.
> > > In the rebuttal, we temporarily retained FPR = 0.05 only to keep consistency with the referenced results in the main paper and avoid introducing confusion or new variables for other reviewers.
> > >
> > > #### 3. PSNR values above 30 dB are generally considered visually indistinguishable.
> > >
> > > We truly appreciate your valuable insight on this point. We will remove this sentence in the revised paper.
> > > As you suggested, we will clearly acknowledge that visual artefacts do exist, and also mention that attackers might be less sensitive to image quality than official watermarking systems are.
> > > We believe this revision will make our argument more balanced and accurate.
> > >
> > >
> > > #### 4. The PSNR of the considered watermarking schemes is also too low.
> > >
> > > We fully acknowledge this point and agree that the low PSNR may partially explain the effectiveness of the attack.
> > > In the revision, we will include examples of watermarking schemes with higher PSNR (e.g., VINE), and we will include the ablation study you suggested.
> > > We agree that this is an important direction and are happy to explore it further.

---

> > > > ### Comment · Reviewer_aB9i · 2025-08-04
> > > >
> > > > There is a misunderstanding. I have read the other reviews and the associated rebuttal. I've found it quite interesting.
> > > >
> > > > ** Can you summarize what you commit to modify in the paper and appendices?**
> > > >
> > > > I understand it encompasses the three points I am very concerned about. But what about other modifications? Please read carefully your rebuttals (not just me, but all the reviewers) and make a list of your commitments. Be as precise as possible.

---

> > > > > ### Author Response · Authors · 2025-08-04
> > > > >
> > > > > **We thank all the reviewers for pointing out several issues. To address these, we commit to the following specific revisions:**
> > > > >
> > > > > ### Terminology and Claims.
> > > > >
> > > > > - We will delete the statements in Lines 555–558 that describe semantic watermarking as a limitation, as they may be misleading.
> > > > >
> > > > > - We will add some discussion of future research directions in the Limitations section, particularly regarding the types of queries that could be helpful for improving the attack, as suggested by Reviewer 1Fpt.
> > > > >
> > > > > - We will cite and briefly discuss relevant works on copy attacks and their corresponding defenses that were proposed in the 2000s, which were previously omitted.
> > > > >
> > > > > - We will replace all occurrences of “black-box” and “no-box” with the appropriate terms such as WOA (Watermarked-Only Attack), KOA (Known-Original Attack), and KMA (Known Message Attack), depending on the context.
> > > > >
> > > > > - We will correct the phrasing in Line 261 and Line 456 by replacing “classical approaches” with “non-learning based methods” to provide a more precise description of the techniques referenced.
> > > > >
> > > > > - We will delete the sentence “PSNR values above 30 dB are generally considered visually indistinguishable”.Instead, we will clearly acknowledge that visual artefacts may still be present even at higher PSNR levels, but also argue that an attacker might not be as keen on the quality as the "official watermarker."
> > > > >
> > > > > - We will revise the equation in line 137 to $x^w = x + w(x)$ in Section 4.1.
> > > > >
> > > > > - We will revise Line 46 to clarify the cause of the baseline's limited performance. Specifically, we will replace Line 46 with:  “The limited performance of the baseline may result from its assumption that the watermark signal is constant across all images. In the WOA setting, its coarse approximation is further constrained by the domain gap between natural images from ImageNet and the clean counterparts of the watermarked images.”
> > > > >
> > > > >
> > > > > ### Additional Implementation Details.
> > > > >
> > > > > - We will report the original PSNR values for each watermarking scheme we used in Section 5 (Watermarking Schemes subsection), to clarify the perceptual quality of the watermarked images.
> > > > >
> > > > > - In Appendix G, we will state the training cost of our attack: “The cost of performing our attack is also modest: for 256 × 256 images, training our attack model takes roughly 40 A100-GPU hours.”
> > > > >
> > > > > - In Appendix C.2, we will include the exact prompts used for attacking the Amazon watermark.
> > > > >
> > > > > - We will also move the statement regarding the length of the embedded watermark message from the appendix to the main paper (Section 5).
> > > > >
> > > > > ### Additional Experiments and Result Updates.
> > > > >
> > > > > - We will correct the results reported for TreeRing in Table 6, add a comparison with Müller’s method, and revise the corresponding description in Appendix E.1 to reflect the correct experimental findings.
> > > > >
> > > > > - We will include the experiment on how the size of the watermarked image set affects attack performance, and present the corresponding result table in Appendix G.
> > > > >
> > > > > - We will add evaluation results for a new watermarking method, VINE, into Table 1, Table 3, and Figure 5.
> > > > >
> > > > > - We will update the FPR threshold used in all experiments from 0.05 to **$10^{-6}$**, as suggested. Accordingly, we will revise the reported results in both the main paper and the Appendix.
> > > > >
> > > > > - We will add an ablation study in Appendix F to investigate the impact of the original PSNR and the number of watermarked images (k) on the performance of our attack.
> > > > >
> > > > >
> > > > > **If any aspect of this correction requires further clarification, we would be happy to elaborate in the revised version.**

---

### Official Review · Reviewer_JXkv · 2025-06-29

**Clarity:** 4
**Significance:** 4
**Originality:** 4
**Rating:** 5
**Confidence:** 4

**Summary:**

The paper presents WMCopier, a no-box watermark-forgery attack that requires no knowledge of, nor access to, the target watermarking algorithm. The attacker merely gathers already-watermarked images produced by a generative-AI service. WMCopier proceeds in three stages: 1. Watermark estimation (Train an unconditional diffusion model on the scraped watermarked images, leveraging the systematic prediction bias that watermark signals add to the model’s noise estimates); 2. Watermark injection (Perform a shallow DDIM inversion (≈ 40 steps of 100) so the latent retains semantic content of any clean image, then run the forward denoising trajectory to let the learned bias imprint the watermark); 3. Refinement (Iteratively optimize the forged image with a score-matching term and an MSE term to reduce artifacts while preserving fidelity).

**Questions:**

See Weaknesses. If my concerns are thoroughly addressed, I would be pleased to further increase my score. Conversely, if the evaluation remains incomplete, I may lower the score accordingly.

**Ethical Concerns:**

["NO or VERY MINOR ethics concerns only"]

**Final Justification:**

Thank you very much for your effort and thorough response.

I have carefully read all the review comments as well as the authors’ responses. I believe the authors have adequately addressed my concerns, and I have raised my score.

**Limitations:**

yes

**Quality:**

4

**Strengths And Weaknesses:**

## Strengths
1. It is a no-box attack.
2. The method is simple and effective.

## Weaknesses
1. The watermarking techniques used to be forged are outdated, particularly the post-processing methods, which were developed five years ago (e.g., Hidden 2018, RivaGan 2019). It remains uncertain whether WMCopier can effectively counter modern post-processing watermarks, such as TrustMark (2023) [1] and VINE (2025) [2]. To ensure a more comprehensive evaluation, experiments involving at least VINE should be included in Tables 1 and 3, as well as Figure 5.

2. How long is the message in the test setting? Is it 32 bits? The ablation study on message pool sizes does not specify bit length.

3. The method requires training one unconditional diffusion model for forging each watermarking method, which could be expensive compared with the baseline [3]. How many A100 GPU hours are required for forging one watermarking model?

4. The training dataset consists of 5,000 images, which is smaller than the amount typically required to train a standard diffusion model. As noted by the authors, this limited size may lead to overfitting and memorization. Is the test set distinct from the training set? How well does this method generalize? I was unable to find these details in the paper, and I’m unsure if I overlooked them.

[1] [TrustMark: Universal Watermarking for Arbitrary Resolution Images](https://github.com/adobe/trustmark)

[2] [Robust Watermarking Using Generative Priors Against Image Editing: From Benchmarking to Advances](https://github.com/Shilin-LU/VINE)

[3] Can simple averaging defeat modern watermarks?

---

> ### Author Rebuttal · Authors · 2025-07-31
>
> Thank you for your insightful comments and constructive feedback. We address your questions and concerns point by point below.
>
> #### **W1.**
> It remains uncertain whether WMCopier can effectively counter modern post-processing watermarks, such as TrustMark (2023) [1] and VINE (2025) [2]. To ensure a more comprehensive evaluation, experiments involving at least VINE should be included...
> #### A1.
> Thank you very much for pointing out several newly proposed watermarking algorithms.
>
> We are also very interested in evaluating whether our method remains effective against these new watermarking schemes.
> Due to time constraints, we have only tested the effectiveness of our attack against the VINE (2025).
> The results are provided below, and we will include them (Table 1,Table 3,and Figure 5) in the updated version of our paper.
>
> | Watermark Scheme | Dataset     | PSNR ↑ | Forged Bit-Accuracy ↑ | FPR@0.05 ↑ |
> |------------------|-------------|--------|------------------------|-------------|
> | VINE             | MS-COCO     | 30.07  | 73.81%                 | 72.90%      |
> |                  | CelebAHQ    | 29.21  | 69.43%                 | 68.80%      |
> |                  | ImageNet    | 31.08  | 71.13%                 | 69.10%      |
> |                  | DiffusionDB | 31.19  | 69.93%                 | 68.90%      |
>
> Our results on VINE were not as strong as those on other watermarking benchmarks.
> We attribute this to the fact that VINE is based on SDXL-Turbo, whose model size is significantly larger than that of typical watermarking algorithms. This enables it to embed longer watermarks with stronger robustness.
> As a result, our relatively lightweight diffusion model may struggle to approximate or replicate such complex watermark embeddings.
> We are considering employing more advanced diffusion architectures with greater capacity, which we leave for future work.
>
> #### **W2.**
> How long is the message in the test setting? Is it 32 bits? The ablation study on message pool sizes does not specify bit length.
> #### **A2.**
> Thank you for the question, and sorry for the confusion.
>
> In our test setting, we follow the default message lengths of each watermarking scheme: 32 bits for DWT-DCT and RivaGAN, 30 bits for HiddeN, and 48 bits for Stable Signature.
> For the ablation study on message pool sizes, we used RivaGAN, whose default message length is 32 bits.
>
> #### **W3.**
> The method requires training one unconditional diffusion model for forging each watermarking method, which could be expensive compared with the baseline [3]. How many A100 GPU hours are required for forging one watermarking model?
> #### **A3.**
> Thank you for you comment.
>
> Our method is indeed more expensive compared to baseline(a simple averaging based approach).
> However, baseline is largely ineffective in practice, achieving only around 11% average success rate and thus posing little real threat to deployed watermarking schemes.
> In contrast, our approach achieves strong performance using a minimal diffusion architecture. For 256×256 images, training the model requires approximately 40 A100 GPU hours.
> Given the potential value an adversary could gain by forging a provider’s watermark, this one‑time cost remains relatively modest.
>
> #### **W4.**
> Is the test set distinct from the training set? How well does this method generalize?
> #### **A4.**
> Sorry for the confusion.
> 1) We have considered the generalization issue in our main experiments by ensuring that the training and testing data come from different distributions.
> Specifically, we train the forgery model on AI-generated images from the DiffusionDB training set, and test it on both the DiffusionDB test set and three real-world datasets (MS-COCO, ImageNet, and CelebA-HQ) that differ significantly in distribution.
> Please see Section 5 for detailed dataset settings.
>
> 2) As shown in Table 1, our method maintains high PSNR and forged bit accuracy even when evaluated on test sets that are significantly different from the training data (e.g., human faces in CelebA-HQ),
> demonstrating its strong generalization ability.

---

> > ### Comment · Reviewer_JXkv · 2025-08-01
> >
> > Thank you very much for your effort and thorough response. I can easily imagine that conducting additional experiments must have been quite challenging, and I truly appreciate your dedication.
> >
> > I have carefully read all the review comments as well as the authors’ responses. I believe the authors have adequately addressed my concerns, and I am inclined to raise my score. I would also like to observe the ongoing discussions between the authors and the other reviewers before making a final decision.
> >
> > Thank you again.

---

> > > ### Author Response · Authors · 2025-08-05
> > >
> > > Thank you very much for your kind words and thoughtful consideration.
> > > We greatly appreciate your time in reviewing our paper and your willingness to raise your score.
> > > Please feel free to reach out if any further clarification or discussion is helpful.

---

### Official Review · Reviewer_1Fpt · 2025-07-02

**Clarity:** 3
**Significance:** 3
**Originality:** 3
**Rating:** 5
**Confidence:** 3

**Summary:**

This paper shows a forgery attack on image watermarks. That is, the attack takes a dataset of watermarked images, and uses an unconditional diffusion model to embed a watermark in a clean image of the attacker’s choice. This attack is “no-box” in that it requires no knowledge of the watermarking scheme being used, nor query access to a watermarking algorithm or detector. It uses only the dataset of watermarked images. The paper includes a large suite of experiments showing effectiveness of the attack. In particular, it shows that the attack is successful at forging Amazon’s closed-source watermark, which is verifiable using a public detection API.

The attack involves training a diffusion model to learn the watermark. The attacker first maps a clean image to its latent representation, then uses the watermark-incorporated diffusion model to denoise it. This denoising process may introduce some artifacts into the clean image, and the paper introduces several optimizations to minimize this distortion. The attacked images are visually very similar to the original clean images.

**Questions:**

- Section 2.2 suggests a bit that all image watermarks involve embedding a fixed message. As mentioned above, certain schemes such as the PRC watermark vary the embedded message. It would be helpful to mention that the class of “fixed-message” watermarks considered in this work is in fact a restricted subclass of schemes.
- How does the attack success vary with the size of the dataset of watermarked images?
- The attack is quite powerful in that it operates in the no-box setting. While query access to a watermarked image generator would help the attacker in theory, it’s not clear to me how the attacker can actually use this access. In developing your attack, did you gain any insight on what kinds of queries would be helpful for the attacker to make in that setting?

**Ethical Concerns:**

["NO or VERY MINOR ethics concerns only"]

**Final Justification:**

Overall, this is a clearly written paper that shows a meaningful attack with comprehensive experiments backing it up. It shows real-world impact by forging Amazon's watermark. The authors satisfactorily addressed my questions in the rebuttal.

While the attack does not fully extend to schemes that vary the message, the paper does consider this case and the authors are running further experiments for the PRC watermark. While I am doubtful that the attack will work against the PRC watermark, the attack against fixed-message schemes is already a significant contribution. I recommend acceptance.

**Limitations:**

Yes

**Quality:**

3

**Strengths And Weaknesses:**

Strengths
- This paper has comprehensive experiments showing the effectiveness of the attack on several different watermarking schemes, and it also includes a comparison to the attack of Yang et al., which it outperforms.
- The paper shows real-world impact of the attack, by forging Amazon’s closed-source watermark.
- The paper is very clearly written, and includes potential defenses as well as the attack.

Weaknesses
- The attack seems to apply only to image watermarks that work by embedding a fixed message in every image. Some image watermarks, such as the PRC watermark, instead vary this message. It is not clear whether the attack would work here.

Overall, this is a clearly written paper that shows a meaningful attack with comprehensive experiments backing it up.

---

> ### Author Rebuttal · Authors · 2025-07-31
>
> We thank you for your positive feedback and we have carefully addressed the raised concerns in the responses below.
>
> #### **W1&Q1.**
> Some image watermarks, such as the PRC watermark, instead vary this message. It is not clear whether the attack would work here.
>
> #### **A1.**
> Thank you for your comment.
> We are still conducting experiments on the PRC watermark and will update the results as soon as they are available.
> In the meantime, we have performed similar experiments on watermarking schemes with varied messages as a potential defense, as described in Appendix E.2.
> In particular, we varied the watermark message over 10, 50, and 100 values.
> We found the success rate of our attack decreased as message diversity increased, dropping to approximately 87%, 77%, and 75%, respectively.
> However, we also found that increasing the number of watermarked images in the training set significantly mitigates this performance drop.
> For example, when the number of watermarked samples was increased to 20,000, the attack success rate recovered to around 90%.
> Please refer to Tables 7 and  8 for detailed results.
> We hope this helps address your concern.
>
> #### **Q2.**
> How does the attack success vary with the size of the dataset of watermarked images?
>
> #### **A2.**
> Thank you for raising this important point.
>
> We evaluated our method using watermark datasets of 1,000, 5,000, and 10,000 images, as shown in the results below (with dataset sizes indicated in parentheses). We will add it in our revision.
>
> | Watermark Scheme |Dataset|PSNR(1k)|Forged bitacc.(1k)|FPR@0.05(1k)|PSNR(5k)|Forged bitacc.(5k)|FPR@0.05(5k)|PSNR(10k)|Forged bitacc.(10k)|FPR@0.05(10k)|
> |-|-|-|-|-|-|-|-|-|-|-|
> | RivaGan |MS-COCO|34.16|81.82%|86.80%|34.07|95.74%|100.00%|34.47|97.81%|100.00%|
> |         |CelebAHQ|35.74|89.10%|98.70%|35.28|98.61%|100.00%|35.25|98.63%|99.80%|
> |         |ImageNet|34.10|81.25%|83.70%|33.87|93.83%|100.00%|34.29|93.53%|95.60%|
> |         |Diffusiondb|34.77|74.76%|67.30%|34.50|90.43%|97.40%|34.96|91.70%|98.60%|
>
>
> #### **Q3.**
> While query access to a watermarked image generator would help the attacker in theory, it’s not clear to me how the attacker can actually use this access. In developing your attack, did you gain any insight on what kinds of queries would be helpful for the attacker to make in that setting?
>
> #### **A3.**
> Thank you for the insightful question.
>
> The most direct way to access watermarked images is to use the image generation service provided by the AI company.
> For example, in our attack on Amazon's watermarking scheme, we used their product Titan text-to-image model, in the same way as a normal user.
> The detailed setup is described in Appendix C.2.
>
> Based on our empirical observations, using a diverse set of prompts instead of repeatedly using a single prompt tends to yield better attack performance.
> This is likely because diverse image content helps the diffusion model better isolate the watermark signal, which remains relatively consistent across samples.
> We are currently further exploring effective query strategies, and we believe this represents an important and promising direction for future research.

---

> > ### Comment · Reviewer_1Fpt · 2025-08-05
> >
> > Thank you for your helpful responses.

---

### Official Review · Reviewer_VmnM · 2025-07-06

**Clarity:** 3
**Significance:** 3
**Originality:** 2
**Rating:** 4
**Confidence:** 4

**Summary:**

This paper introduces WMCopier, a novel no-box watermark forgery attack targeting invisible watermarks in images generated by AI systems. Unlike prior methods requiring access to watermarking algorithms or paired data, WMCopier trains an unconditional diffusion model on collected watermarked images to estimate the watermark distribution. It then injects the estimated watermark into arbitrary clean images through a shallow inversion and refinement process, producing forged images that convincingly mimic genuine watermarked content. The authors demonstrate that WMCopier achieves high forgery success rates across multiple open-source and closed-source watermarking schemes, including Amazon’s system, while maintaining strong visual fidelity. The paper also explores potential defenses and analyzes the robustness of the forged watermarks.

**Questions:**

Please see the weaknesses.

**Ethical Concerns:**

["NO or VERY MINOR ethics concerns only"]

**Final Justification:**

The authors have addressed several key concerns raised in the initial review. They clarified the training/testing split, corrected the TreeRing evaluation, and provided a fair comparison with Müller et al. While Müller’s method is slower, it avoids retraining and large watermarked datasets—offering practical advantages not discussed in depth in the original submission. Some issues remain partially addressed, such as scalability to higher resolutions and comparisons with other semantic watermarking methods like Gaussian Shading. However, the core contributions are now better supported. Given these improvements and the consensus among reviewers, I have raised my score to borderline accept.

**Limitations:**

Yes

**Paper Formatting Concerns:**

The supplementary material is appended directly after the main paper in the same PDF file, rather than provided as a separate file.

**Quality:**

3

**Strengths And Weaknesses:**

$\textbf{Strengths:}$

1 -- The paper presents a novel no-box watermark forgery attack that requires no prior knowledge of the watermarking scheme, addressing a realistic and challenging threat model.

2 -- Extensive experiments demonstrate state-of-the-art performance across multiple watermarking schemes, including a deployed commercial system (Amazon Titan).

3 -- The method is well-motivated and clearly presented, with thoughtful exploration of potential defenses and limitations, contributing valuable insights for improving watermark robustness.

$\textbf{Weaknesses:}$

1 -- The paper does not compare WMCopier with semantic watermarking approaches like Tree-Ring and Gaussian Shading; these methods are now well-established rather than as claimed “early-stage,” and evaluating against them is essential given their relevance to forging semantic watermarks.

2 -- It is unclear whether the proposed method requires training a separate diffusion model for each watermarking scheme or if a single diffusion model can generalize across multiple watermarks.

3 -- As shown in the paper (Table 3), the forged watermarks exhibit significantly lower robustness to common image distortions than genuine ones, making them easy to break or reveal through routine post-processing and undermining the attack’s practicality in real-world use.

4 -- The authors acknowledge in Appendix G that training on limited auxiliary datasets may cause diffusion models to memorize patterns, leading to potential overfitting. However, the paper lacks systematic evaluation of how this overfitting affects generalizability to images outside the training distribution, casting doubt on the claimed robustness in realistic scenarios.

5 -- The authors should discuss or compare their approach with Black-Box Forgery Attacks on Semantic Watermarks for Diffusion Models (Müller et al., 2024-- CVPR2025), which effectively forges semantic watermarks in a black-box setting; such a comparison is necessary to contextualize WMCopier’s strengths and limitations

---

> ### Author Rebuttal · Authors · 2025-07-31
>
> Thank you very much for your valuable comments. Below, we carefully address each of your concerns in detail.
>
> #### **W1&W5.**
> 1) The paper does not compare WMCopier with semantic watermarking approaches and compare their approach with Black-Box Forgery Attacks on Semantic Watermarks for Diffusion Models (Müller et al., 2024-- CVPR2025), which effectively forges semantic watermarks in a black-box settings...such a comparison is necessary to contextualize WMCopier’s strengths and limitations.
>
> 2) Semantic watermarking is now well-established rather than as claimed “early-stage”.
> #### **A1.**
> Thank you for raising this issue.
> 1) We previously evaluated attacks against the semantic watermarking method Treering, as discussed in Appendix E.1.
>     At that time, we were unable to attack it successfully. Upon revisiting our experiments, we realized that this was due to a mistake. We had overlooked that Treering uses a different definition of the true positive rate (TPR).
>     After correcting this issue, we found that our attack can successfully break the Treering watermark without any parameter tuning.
>     The FPR threshold is set according to the default used in the Treering paper, and we will release the updated model checkpoints on Treering soon.
>     Our attack results and the comparison with Müller’s black-box attack are as follows:
>
>     | Watermark Scheme | Dataset     | PSNR ↑ (Müller) | FPR@0.01 ↑ (Müller) | PSNR ↑ (Ours) | FPR@0.01 ↑ (Ours) |
>     |------------------|-------------|------------------|----------------------|----------------|--------------------|
>     | Treering         | MS-COCO     | 26.14            | 100.00%              | 32.72          | 100.00%            |
>     |                  | CelebAHQ    | 25.22            | 100.00%              | 31.52          | 100.00%            |
>     |                  | ImageNet    | 26.82            | 100.00%              | 32.99          | 100.00%            |
>     |                  | Diffusiondb | 25.19            | 100.00%              | 32.78          | 100.00%            |
>
>     As shown above, our method achieves a consistently higher image quality while maintaining the same success rate(FPR).
>
>     Their method is primarily tailored for semantic watermarking and has not been validated on invisible watermarking schemes, making its general applicability unclear. In contrast, our method has demonstrated effectiveness on both semantic and invisible watermarks. While their approach is relatively lightweight in terms of computation, our WMCopier involves training a diffusion model, which introduces some computational cost. However, this cost is modest (e.g., 40 A100 GPU hours for 256×256 images), and enables broader applicability and higher success rates across a range of watermarking schemes.
>
> 2) Regarding the use of the term "early-stage," our intention was to refer to watermarking methods that have emerged in the past two to three years and are still relatively few in number. We appreciate you pointing this out and will revise the wording to use a more accurate and appropriate expression.
>
> #### **W2.**
> It is unclear whether the proposed method requires training a separate diffusion model for each watermarking scheme.
>
> #### **A2.**
> Thank you for your comment.
>
> Our approach does require training a separate diffusion model for each watermarking scheme.
> However, as discussed in our threat model, we assume the reasonable scenario that service providers do not frequently change the watermarking scheme in practice.
> The cost of performing our attack is also modest: for 256 × 256 images, training our attack model takes roughly 40 A100‑GPU hours.
> Given the potential value an adversary could gain by forging a provider’s watermark, this one‑time cost remains relatively modest.
>
> #### **W3.**
> As shown in the paper (Table 3), the forged watermarks exhibit significantly lower robustness to common image distortions than genuine ones, making them easy to break or reveal through routine post-processing and undermining the attack’s practicality in real-world use.
>
> #### **A3.**
> Thank you for your comment.
>
> We agree that forged watermarks sometimes achieve slightly lower bit‑accuracy than genuine ones after common image distortions; in our experiments, this gap is typically less than 10 percentage points.
> However, such a small gap is difficult to exploit in practice. Any detection threshold low enough to catch most forged images would likely also reject many genuine watermarked images that have undergone compression or noise, thereby reducing the true positive rate required for robustness.
> For example, a genuine watermark that has been JPEG-compressed may yield a similar bit-accuracy as a forged watermark under the same distortion.
> We further analyze this in Appendix D.2, where we plot ROC curves for distinguishing between genuine and forged watermarks under typical distortions. The resulting AUCs are close to 0.5, indicating that the robustness gap is insufficient for reliably filtering out forgeries.
>
> #### **W4.**
> The paper lacks systematic evaluation of how this overfitting affects generalizability to images outside the training distribution.
>
> #### **A4.**
> Thank you for your comment. This is also something we consider important.
>
> Therefore, we have considered the generalization issue in our main experiments by ensuring that the training and testing data come from different distributions.
> To evaluate generalization, we train the forgery model on AI-generated images from the DiffusionDB training set and test it on both the DiffusionDB test set and three real-world datasets (MS-COCO, ImageNet, and CelebA-HQ) that differ significantly in distribution.
> Please see Section 5 for detailed dataset settings.
> Thanks to our shallow diffusion design and subsequent refinement step, our method performs consistently well across all datasets as shown in Table 1 of the main paper.

---

> > ### Comment · Reviewer_VmnM · 2025-08-05
> >
> > Thank you for the additional results and clarifications. A few brief follow-ups:
> >
> > -- If the model is trained on DiffusionDB and evaluated on MS-COCO, ImageNet, and CelebA-HQ as stated in the rebuttal, it would help to clearly mention this setup in the caption of Table 1 or in the main text, as this detail is not currently obvious.
> >
> > -- Regarding TreeRing, it is understandable that a metric mismatch led to confusion, though this should have been identified earlier given the importance of the baseline. In light of this, have the authors also evaluated on Gaussian Shading? It may reveal similar issues or trends.
> >
> > -- Lastly, for the comparison with Müller et al., could the authors clarify which generative model was used? Their reported PSNR is closer to 30 dB, while the values presented here are lower (~26 dB), and this discrepancy may reflect differences in setup.

---

> ### Comment · Area_Chair_9cPq · 2025-08-05
>
> Dear reviewer,
>
> Please read the rebuttal and start discussion with authors.
>
> AC

---

> ### Author Response · Authors · 2025-08-06
>
> Thank you for your detailed and constructive feedback. We have carefully addressed your concerns below and hope our responses help clarify the points raised.
>
> ### 1.
> Thank you for your suggestion. In the revised version, we will clearly state the training and evaluation setup in a more prominent location, such as below Table 1, as you suggested.
>
> ### 2.
> We sincerely appreciate your understanding and your thoughtful suggestion.
> As this experiment was initiated after receiving your comments, the available time has been somewhat limited, and we kindly hope for your understanding regarding this timing.
> We are working hard to complete the Gaussian Shading evaluation and will include the results in the revised version if available, or follow up with an update shortly thereafter.
>
> ### 3.
> We would like to clarify that the discrepancy in PSNR originates from an error in Müller et al.’s earlier reporting, not from our evaluation. The authors have since acknowledged this issue in their official GitHub repository and corrected it in a commit dated May 30. You can find their explanation, along with a comparison of the incorrect and corrected PSNR values (from ~30 to ~23), in the associated issue discussion and commit message.
>
> In their CVPR camera-ready version (May 2025), Müller et al. reported a PSNR of ~30. However, in the updated arXiv version (June 2025), the PSNR dropped to around 23–24, which is consistent with our results. We investigated this discrepancy and confirmed that we had used the correct official code with default settings for Stable Diffusion 2.1 base. The earlier PSNR calculation in their code was incorrect and led to inflated results.
>
> We would also like to reiterate the advantages of our method over Müller et al.'s.
> - First, Müller et al.'s method requires approximately 25-40 minutes to forge a single image on an NVIDIA A100 GPU with over 30 GB of memory usage, as reported in their paper.
> To provide a fair comparison, we tested our method under the same hardware and memory constraints.
> In this setting, our model is able to process 10 images in 192 seconds total, achieving significantly higher throughput.
>
> - Second, Müller et al.'s method is specifically tailored for semantic watermarks.
> To evaluate its generalizability, we tested it on invisible watermarking using Stable Signature, which has the same spatial resolution (512×512) as TreeRing.
> Due to the extremely long runtime of their method per image, we randomly sampled 50 images per dataset for evaluation.
> Results show that their method is largely ineffective on invisible watermarks, while our method remains highly effective.
> | Dataset     | PSNR  (Müller) | Forged Bit-Accuracy  (Müller) | FPR@1e-6  (Müller) | PSNR  (Ours) | Forged Bit-Accuracy  (Ours) | FPR@1e-6  (Ours) |
> |-------------|------------------|-------------------------------|---------------------|----------------|-------------------------------|--------------------|
> | MS-COCO     | 25.66            | 45.70%                        | 0.00%               | 31.29          | 98.04%                        | 94.60%             |
> | CelebA-HQ   | 24.73            | 51.23%                        | 0.00%               | 30.54          | 96.04%                        | 100.00%            |
> | ImageNet    | 25.91            | 47.71%                        | 0.00%               | 31.33          | 97.03%                        | 98.60%             |
> | DiffusionDB | 26.12            | 48.45%                        | 0.00%               | 31.59          | 96.24%                        | 96.60%             |

---

> > ### Comment · Reviewer_VmnM · 2025-08-07
> >
> > Thank you for the detailed rebuttal and updated experiments. The clarification on generalization (DiffusionDB → MS-COCO/ImageNet/CelebA-HQ) and the corrected TreeRing evaluation are appreciated.
> >
> > Regarding the comparison with Müller et al., I acknowledge that their method is slower (~30–40 minutes per image). However, it has practical advantages: it does not require retraining a diffusion model for each watermarking method, nor access to a large number of watermarked images. Additionally, it operates on 512×512 resolution, while WMCopier is currently evaluated on 256×256, and scaling up would likely increase training cost for WMCopier. These trade-offs should be more explicitly discussed.
> >
> > That said, the authors have partially addressed my earlier concerns, and in light of the broader reviewer discussion, I am raising my score. For the camera-ready version, I recommend clearly stating the training/testing split in Table 1 or the main text, including the corrected TreeRing results with appropriate metric clarification, and adding a brief discussion on the trade-offs with Müller et al. (e.g., training cost, data requirements, resolution). Adding Gaussian Shading as a relevant semantic watermarking baseline would further strengthen the paper.

---

> > > ### Author Response · Authors · 2025-08-08
> > >
> > > We sincerely thank the reviewer for the encouraging feedback and for raising the score.
> > > We truly appreciate your recognition of the updated experiments, clarification on dataset generalization, and the revised TreeRing results.
> > >
> > > For the revised version, we will explicitly include the following updates as per your helpful suggestions:
> > >
> > > - Clearly state the training/testing split in both Table 1 and the main text, to avoid ambiguity.
> > > - Include the corrected TreeRing results.
> > > - Add a discussion on trade-offs in comparison to Müller et al.
> > > - Incorporate Gaussian Shading as a semantic watermarking baseline to further strengthen the baseline comparisons.
> > >
> > > Thank you again for your constructive and thoughtful comments.

---

### Decision · Program_Chairs · 2025-09-17

**Decision:**

Accept (poster)

**Comment:**

The manuscript proposes ​​WMCopier​​, the first effective no-boxwatermark forgery attack, training a diffusion model on watermarked images to forge target watermarks into arbitrary images without algorithm access. It is novel, effective, and well evaluated. The reviewer's concerns are mostly about missing modern schemes comparison, generalization and some inaccurate terminology. The authors addressed these concerns by adding new results and clearly clarify their limitations. Based on the rebuttal process, I recommend acceptance of this manuscript.